# Anillin directly crosslinks microtubules with actin filaments

Ilina Bareja[1], Ondřej Kučera [2,3], Irene Istúriz Petitjean[1], Beatriz Eugenia Orozco Monroy [1], Jan Sabo [2], Marcus Braun [2], Zdenek Lansky[2], Gijsje H Koenderink [1✉] & Marileen Dogterom [1✉]

## Abstract

**Complex morphogenetic processes such as cell division require a tight coordination of the activities of microtubules and actin filaments. There is evidence that anillin, conventionally known as an actin-binding and -bundling protein, regulates microtubule/actin crosstalk during cell division. However, it is unknown whether anillin binds directly to microtubules and whether it is sufficient to establish crosslinking between microtubules and actin filaments. Here we address both questions by developing an in vitro system for observing anillin-mediated interactions with actin filaments and dynamic microtubules via total internal-reflection fluorescence microscopy. We find that anillin can interact directly with microtubules and promote microtubule bundling. We confirm that anillin binds and bundles actin filaments, and find that it has a strong preference for actin bundles over individual filaments. Moreover, we show that anillin can directly crosslink microtubules and actin filaments, cause sliding of actin filaments on the microtubule lattice, and transport actin filaments by the growing microtubule tip. Our findings indicate that anillin can potentially serve as a direct regulator of microtubule/actin crosstalk, e.g., during cell division.**

**Keywords** Cytoskeleton; Dynamic Instability; Reconstituted System; Cell Division
**Subject Category** Cell Adhesion, Polarity & Cytoskeleton

## Introduction

Anillin is a highly conserved scaffolding protein and actin crosslinker that serves as a central regulator of cytokinesis, being involved in the formation and stabilisation of the cytokinetic ring (Hickson and O'Farrell, 2008; Zhang and Maddox, 2010) as well as in intercellular bridge maturation (Panagiotou et al, 2022). Anillin is primarily in the cell nucleus during interphase, where it has been shown to regulate transcription initiation-related Pol II clustering and target gene expression (Cao et al, 2025). Therefore, it interacts with the cytoskeleton in a cell cycle-dependent manner. However, recent studies have shown that anillin's functions also extend beyond cytokinesis. Anillin participates in neuronal migration during development (Tian et al, 2015; Rehain and Maddox, 2015), in myelination of neurons (Patzig et al, 2016; Erwig et al, 2019), and in cell–cell adherens junctions in epithelial tissues (Reyes et al, 2014; Arnold et al, 2019; Budnar et al, 2019). In these processes, anillin is consistently observed to co-localise with actin filaments, consistent with in vitro studies showing that anillin directly binds filamentous actin (Jananji et al, 2017; Matsuda et al, 2020; Kučera et al, 2021). At the same time, however, microtubules are also critical for these cellular events. For instance, in cell division, coordination between the microtubule-based mitotic spindle and the actin-based contractile ring ensures that the contractile ring assembles precisely in the middle of the cell (Foe and von Dassow, 2008; Yüce et al, 2005; Canman et al, 2008). Conversely, the actin cortex contributes to proper spindle positioning (Kunda and Baum, 2009). Furthermore, microtubules regulate differences in actin cortex contractility between the equator and the poles during division (Tse et al, 2011; van Oostende Triplet et al, 2014) and enable the correct positioning of the cytokinetic furrow (Shannon et al, 2005). Given the reliance of these cellular processes on both actin and microtubules, it is plausible that anillin's functions may involve an association with the microtubule cytoskeleton, either directly or through intermediary mechanisms.

A structural link between the actomyosin ring and spindle microtubules has been suggested through the spindle microtubule-associated protein Drosophila RacGAP (Gregory et al, 2008) and its human counterpart, Ect2 (Frenette et al, 2012). This interaction is proposed to enable anillin to stabilise the division plane by potentially crosslinking spindle microtubules, the contractile ring, and the membrane (Akhshi et al, 2014). At the polar cortex, anillin has been observed to colocalize with astral microtubules in both *C. elegans* embryos and HeLa cells (van Oostende Triplet et al, 2014). Using truncated constructs corresponding to the C-terminal half of anillin, it was shown that both *C. elegans* and human anillin co-sediment with taxol-stabilised microtubules, suggesting a potential direct interaction. In cells, taxol stabilisation was also found to enhance the colocalization of anillin with microtubules. The microtubule-binding site has been narrowed down to a basic region (residues DFEINIE) within the C-terminal half of anillin

[1]Department of Bionanoscience, Kavli Institute of Nanoscience, Faculty of Applied Sciences, TU Delft, Delft, The Netherlands. [2]Institute of Biotechnology of the Czech Academy of Sciences, BIOCEV Research Centre, Prague, Czech Republic. [3]Department of Engineering Technology, South East Technological University, Waterford, Ireland.
✉E-mail: G.H.Koenderink@tudelft.nl; M.Dogterom@tudelft.nl

(van Oostende Triplet et al, 2014). It has also been hypothesized that astral microtubules may facilitate the removal of anillin from the actin cortex at the cell poles, ensuring that actomyosin contraction is restricted to the equatorial cortex to drive furrow ingression during cytokinesis (Tse et al, 2011; Akhshi et al, 2014). These observations support the idea that anillin could act as a crosslinker between F-actin and microtubules. However, the nature of anillin's interaction with microtubules remains poorly understood, and it is unclear whether this interaction affects anillin's ability to bind and bundle F-actin.

Here, we investigate the potential direct interaction between anillin and microtubules and its implications for anillin-mediated crosstalk between F-actin and microtubules. We opted for in vitro reconstitution assays because anillin has many different interaction partners in the cell that make it difficult to selectively study specific cytoskeletal interactions (Naydenov et al, 2021; Beaudet et al, 2020). We used full-length human anillin (isoform 2, 1087 amino acids) in the in vitro reconstitution assays instead of truncated versions to better reflect the in vivo scenario. Using total internal-reflection fluorescence (TIRF) microscopy, we observed and quantified the interactions of anillin with dynamic and stabilised microtubules in the absence and presence of actin filaments. We found that anillin binds directly to microtubules on the stable as well as dynamic lattice and also at the plus end. Further, anillin enables the bundling of microtubules as well as the recruitment of actin filaments onto microtubules, causing actin filaments to slide and bundle on microtubules. Our work shows that anillin can mediate crosstalk between the actin and microtubule cytoskeleton by serving as a crosslinker, acting in a similar manner as previously identified cytolinkers such as ACF7/MACF and Gas2L (Dogterom and Koenderink, 2019; Coles and Bradke, 2015; Pimm and Henty-Ridilla, 2021). Our findings hereby provide new insights into the functional roles of anillin across diverse cellular processes.

# Results

## Anillin binds diffusively to microtubules and affects catastrophes and rescues

Anillin is a large (~124 kDa) multidomain protein with many different known interaction partners (Fig. 1A) (Naydenov et al, 2021; Beaudet et al, 2020). Its N-terminal half contains three distinct domains that, respectively, bind the actin-nucleator protein formin (DIAPH3), F-actin, and non-muscle myosin 2. Its C-terminal half contains the anillin-homology domain, which has binding regions for RhoA, septins, the Rho guanine nucleotide exchange factor Ect2, importins and lipids, followed by a PH domain that binds lipids. The two halves are separated by a spacer region of undefined fold. Sequence analysis indicates that this region contains a coiled-coil domain that likely facilitates anillin multimerisation (schematic representation in Fig. 1B) (Piekny and Maddox, 2010). To test for anillin binding to microtubules, we used an in vitro reconstitution assay in which we grew dynamic microtubules from stabilised GMPCPP seeds on a passivated microscope coverslip (Fig. 1C). Upon the introduction of anillin during TIRF imaging, we observed distinct anillin particles interacting directly with the microtubules. These interactions were characterised by a diffusive binding behaviour, indicating dynamic

association with the microtubule lattice (Fig. 1D–H). Further, we observed a higher intensity of anillin on the stable GMPCPP seeds as compared to the dynamic GDP lattice (Figs. 1D–H and EV1A). A similar binding preference was also observed for taxol-stabilised microtubules (Fig. EV1B). Interestingly, anillin sometimes tracked the growing plus tip of microtubules (white arrowheads in Fig. 1F,H) and/or accumulated on shrinking ends (green arrowheads in Fig. 1F,H), particularly at higher concentrations (see also Fig. EV1C). The observed preference for GMPCPP- or taxol-bound tubulin as well as tip tracking of growing microtubules indicates a preference for GTP tubulin. In about half of the cases where anillin was tip-localised the accumulations were seen only at shrinking ends, while in the other half of the cases anillin localised both at growing and shrinking ends.

On looking closely at the diffusion of anillin on the GDP lattice, we observed that the anillin signal is not homogenous. Intensity profiles for anillin along microtubules showed distinct peaks indicating larger particles, potentially oligomeric species (Appendix Fig. S1A). To test whether these particles are formed already in solution or only on microtubules, we measured the molecular weight distribution of anillin in solution using mass photometry (or iSCAT). We found that at concentrations similar to those used for the TIRF experiments (5, 15 or 50 nM), anillin was present predominantly as a monomer (Appendix Fig. S1B–D). Apparently, anillin forms larger particles only when bound to microtubules. More specifically, we observed these larger anillin particles only on dynamic MTs, i.e., in the presence of GTP tubulin in solution and never on microtubules stabilised with GMPCPP or taxol (Fig. EV1A,B; Appendix Fig. S1A). To test whether this could be a result of anillin binding directly to free tubulin in solution, we also performed iSCAT with anillin and tubulin together. Note that due to the limitations of detection of particles in iSCAT, we could only visualise a total of 100 nM protein (tubulin and anillin combined). We did not see binding under these conditions (Appendix Fig. S1F). However, this interaction cannot be ruled out since the concentration of tubulin used in the TIRF assays is much higher.

To probe the binding kinetics of anillin on single microtubules and compare this for a GDP vs GMPCPP lattice, we used fluorescence recovery after photobleaching (FRAP) experiments, following a similar procedure as reported previously for other actin–microtubule cross-linkers (Preciado López et al, 2014b; van de Willige et al, 2019). As shown in Fig. 1I,J, the fluorescence intensity recovered, but incompletely. To estimate the off-rate $k_{off}$ of anillin from the microtubule lattice, we performed single exponential fitting. The median $k_{off}$ value for anillin was higher on GDP (dynamic) microtubules ($1.07\,s^{-1}$) as compared to GMPCPP microtubules ($0.26\,s^{-1}$), consistent with our observations from TIRF imaging that showed a higher anillin signal on GMCPP microtubules (Fig. 1K).

We next analysed whether anillin affected the kinetic parameters characterising the dynamic instability of microtubules (Fig. 1L–O; Appendix Table S1). In the analysis, we distinguished between microtubules that exhibited pronounced anillin accumulation at the growing end from microtubules that did not. Anillin at a concentration of 100 nM did not cause any major changes in the microtubule growth and shrink rates compared to the control (no anillin) case (Fig. 1L,M, respectively). Anillin increased the catastrophe frequency by multiple fold, but only for microtubules where anillin was end-accumulated (Fig. 1N). Lastly, anillin

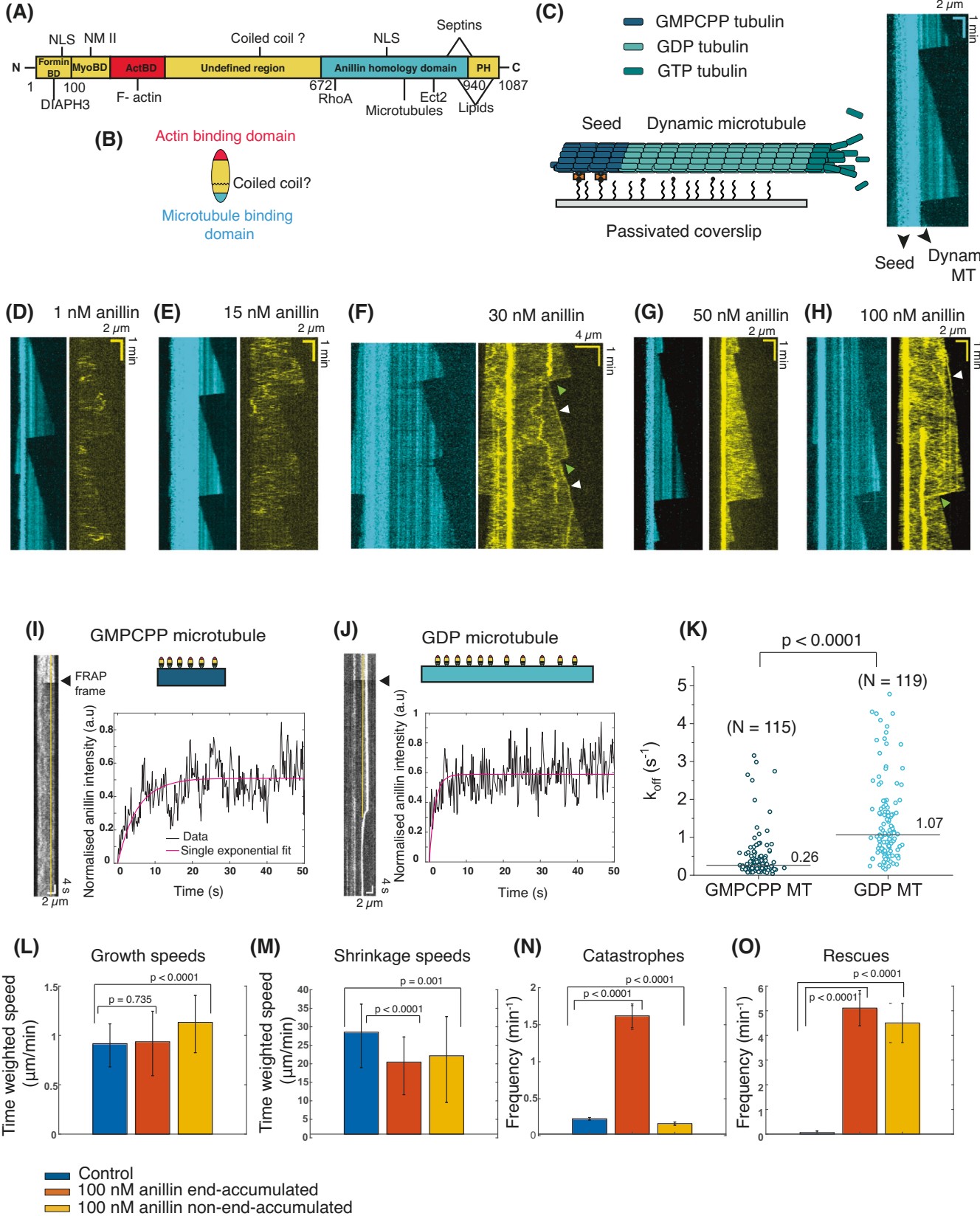

◄ **Figure 1. Anillin as a scaffolding and actin–microtubule crosslinking protein.**

(A) Schematic representation of the domain structure of human anillin isoform 2 (adapted from Naydenov et al, 2021 and Beaudet et al, 2020). Numbers indicate the specific residue numbers mapped for this isoform. Formin BD: formin-binding domain, MyoBD: myosin-binding domain, ActBD: actin-binding domain, PH: Pleckstrin homology domain. Lines indicate recognition sites for anillin's binding partners. NLS: Nuclear localisation signal, DIAPH3: Diaphanous Related Formin 3, NMII: Non-muscle myosin II. (B) Schematic representation of anillin, indicating its F-actin-binding domain and putative microtubule-binding site and intervening coiled-coil domain. (C) Left: Schematic of the TIRF assay to test binding of anillin to dynamic microtubules grown from stable GMPCPP microtubule seeds tethered to the surface using neutravidin-biotin linkages. Right: Kymograph of a control dynamic microtubule (without anillin; cyan, weaker intensity) growing from stable microtubule seeds (cyan, higher intensity). (D–K) Anillin directly binds to microtubules. (D–H) Example kymographs of anillin (yellow) bound to and diffusing on dynamic microtubules (cyan, weaker intensity) growing from stable microtubule seeds (cyan, higher intensity) at different anillin concentrations of 1 nM (D), 15 nM (E), 30 nM (F), 50 nM (G) and 100 nM (H). White arrowheads point to the accumulation of anillin on growing microtubule ends, while green arrowheads point to accumulations of anillin on shrinking microtubule ends. (I, J) Kymographs and corresponding recovery curves after photobleaching of anillin on a GMPCPP microtubule and GDP microtubule, respectively (see schematics). A line is drawn (5-pixel wide) on the kymograph (shown in yellow) to obtain the intensity profiles. For the GDP microtubule, the line is drawn such that it does not pass through large particles. Normalisation was done by first subtracting the fluorescence intensities from the minimum intensity after bleaching and then dividing by the maximum intensity before bleaching, which was taken as the average of the values pre-FRAP. Pink solid curves show single exponential fits. Respective $R^2$ values for (I, J): 0.2875, 0.1481. (K) Off-rates ($k_{off}$) obtained from single exponential fits of recovery curves for anillin bound to GMPCPP or GDP microtubules. The horizontal lines represent the median values. Exact $P = 2.48 \times 10^{-25}$ using a Mann–Whitney $U$ test. (L–O) Effect of anillin on microtubule dynamics. $N = 186$, 32 and 124 events for control, 100 nM end-accumulated anillin and 100 nM non-end-accumulated anillin, respectively in (L); 113, 69 and 48 events for control, 100 nM end-accumulated anillin and 100 nM non-end-accumulated anillin, respectively in (M); 134, 99 and 44 events for control, 100 nM end-accumulated anillin and 100 nM non-end-accumulated anillin, respectively in (N); 1, 50 and 32 events for control, 100 nM end-accumulated anillin and 100 nM non-end-accumulated anillin, respectively in (O). For (L, M), the error bars represent the weighted standard deviation; heights of the bars represent the mean. For (N, O), the error bars are the frequency divided by the square root of the number of events; heights of the bars represent the mean. Using a two-sample Student's $t$ test (with Welch's correction): For (L), $P = 0.735$ and $7.46 \times 10^{-11}$ for control vs end-accumulated and non-accumulated anillin, respectively. For (M), $P = 9.48 \times 10^{-10}$ and 0.001 for control vs end-accumulated and non-accumulated anillin, respectively. For (N), $P = 9.92 \times 10^{-96}$ for control vs end-accumulated and $1.31 \times 10^{-31}$ control vs non-accumulated anillin. For (O), $P \approx 0$ for both control vs end-accumulated and non-accumulated anillin. Source data are available online for this figure.

strongly enhanced the rescue frequencies, both for microtubules with and without accumulations of anillin at the plus end (Fig. 1O).

## Anillin can crosslink microtubules

Knowing that anillin bundles actin filaments (Kučera et al, 2021) and having just demonstrated that it binds to microtubules, we wondered whether it might also have the ability to bundle microtubules. To test this possibility, we first immobilised biotinylated GMPCPP microtubule seeds and next added a solution containing non-biotinylated seeds together with 10 nM anillin. As shown in Fig. 2A, we observed recruitment of non-biotinylated microtubules (magenta) on the surface-immobilised (biotinylated) microtubules (cyan), accompanied by an increase in the anillin intensity in the region of an overlap (yellow). In a different experiment, we added 12 nM anillin to a channel which only contained surface-immobilised GMPCPP microtubule seeds (using tubulin antibodies). We observed that microtubules that crossed each other tended to zipper together upon addition of anillin (Fig. 2B; Movie EV1). The force of the crosslinking was high enough to detach microtubules from the antibodies and occasionally break the microtubule (position of breaking is shown by a yellow arrow in Fig. 2B, see also Movie EV2). These results demonstrate that anillin crosslinks microtubules. The off-rate of anillin from microtubule-microtubule overlaps as determined by FRAP measurements was comparable to that on single GMPCPP microtubules (Fig. 2C,D). We also tested whether the microtubule crosslinking by anillin depends on the polarity of microtubules using IRM combined with TIRF microscopy and observed that it is polarity invariant (Appendix Fig. S2).

## Anillin binds and bundles actin filaments

In order to test whether anillin binds F-actin under the conditions we used for the microtubule assay (and later for the actin–microtubule crosstalk assay), we performed a TIRF assay

where we immobilised single biotinylated actin filaments on the coverslip using neutravidin in MRB80 buffer (Fig. 3A). When we added 10 nM anillin, we observed distinct anillin particles binding and unbinding from the filaments (still images and kymograph in Fig. 3A, left). The residence times ranged between 3 and 47.3 s with a median of 9.5 s (Fig. 3A, right). At a higher concentration (100 nM) of anillin, we observed a similar, inhomogeneous anillin decoration of the actin filaments (Fig. EV2A). When anillin was instead mixed together with non-biotinylated actin filaments in solution and injected into the channel, we observed anillin-mediated F-actin bundling. At lower concentrations (5 nM) of anillin, we could image the bundling process (Fig. 3B). We observed increased recruitment of anillin after the bundling was initiated, indicating a preference of anillin for actin bundles over single actin filaments. At higher concentrations (100 nM) of anillin, we observed many thick and relatively static bundles already formed before image acquisitions were started (Fig. EV2B). Consistent with stronger binding to actin bundles as compared to single filaments, FRAP assays revealed a somewhat higher median off-rate (1.06 s$^{-1}$) for anillin on individual actin filaments as compared to anillin on actin bundles (0.73 s$^{-1}$) (Fig. 3C,D,E).

To visualise the bundles with greater resolution, we then used negative-stain electron microscopy. In a given field of view, we observed F-actin bundles of different widths in the presence of 100 nM anillin, and these widths could also vary within the same bundle (Fig. 3F, right), as compared to individual actin filaments in the absence of anillin (Fig. 3F, left). At a higher magnification, we also observed that these bundles were quite loose, with the density of anillin being inhomogeneous on different actin filaments within a bundle (Figs. 3F and EV2C).

## Anillin crosslinks actin filaments and microtubules

Having established that anillin can bind microtubules as well as F-actin, we next tested whether anillin can mediate crosslinking

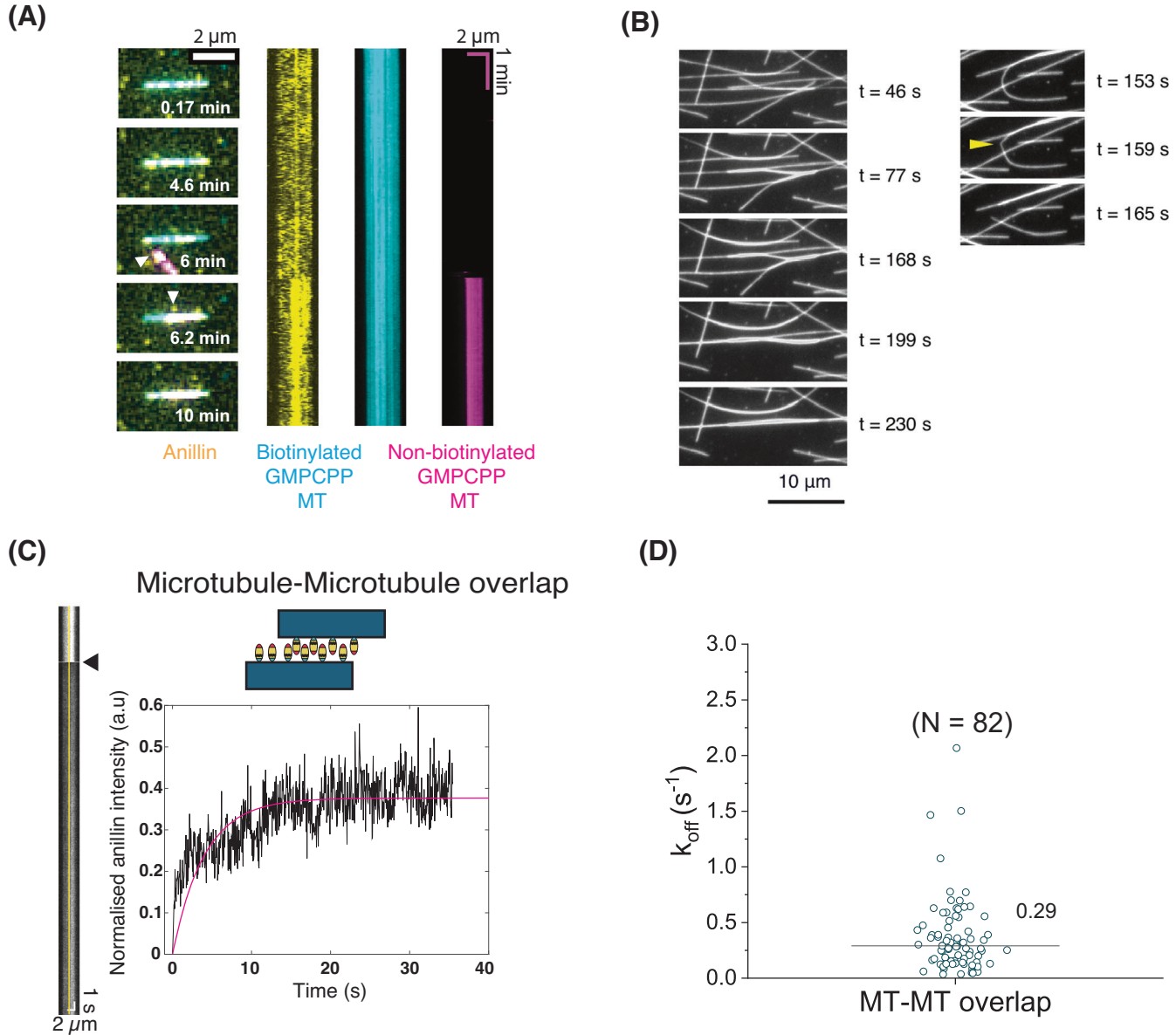

**Figure 2. Anillin can crosslink microtubules.**

(A) Time-lapse image series (left) and corresponding kymographs (right) showing the recruitment of a non-biotinylated GMPCPP microtubule (magenta, white arrowhead) to a surface-immobilised (biotinylated) GMPCPP microtubule (cyan) by anillin (10 nM, yellow). (B) Left: Time-lapse image series showing GMPCPP microtubules zippering together after flowing in anillin (12 nM) at $t = 0$. Right: Time-lapse image series showing the detachment and breaking of a GMPCPP microtubule while being crosslinked to another GMPCPP microtubule upon addition of 12 nM anillin. (C) Kymograph and corresponding recovery curve after photobleaching of anillin on a microtubule-microtubule overlap. A line is drawn (5-pixel wide) on the kymograph (shown in yellow) to obtain the intensity profiles. Normalisation was done by first subtracting the fluorescence intensities from the minimum intensity after bleaching and then dividing by the maximum intensity before bleaching, which was taken as the average of the values pre-FRAP. Pink solid curve shows the single exponential fit ($R^2$ value 0.4794). (D) Off-rates ($k_{off}$) obtained from single exponential fits of recovery curves for anillin bound to microtubule-microtubule overlaps. The horizontal line represents the median value. Source data are available online for this figure.

between stabilised microtubules and actin filaments. As a control, we first imaged a channel containing surface-immobilised GMPCPP microtubules and free-floating, phalloidin-stabilised actin filaments (Fig. EV3A, top). We observed no co-localisation of the two polymers under these conditions, showing that actin filaments and microtubules do not interact. Once 12 nM anillin was flown into this channel, actin filaments were rapidly recruited onto the microtubules (Fig. EV3A, bottom). Also when anillin and actin filaments were flown together into a channel containing surface-immobilised GMPCPP microtubules, actin filaments were recruited onto the microtubules by anillin (Fig. EV3B). These findings show that anillin can indeed form crosslinks between microtubules and actin filaments.

When short actin filaments were introduced in the presence of anillin into a channel with long surface-immobilised microtubules, multiple actin filaments could be attached to and diffuse

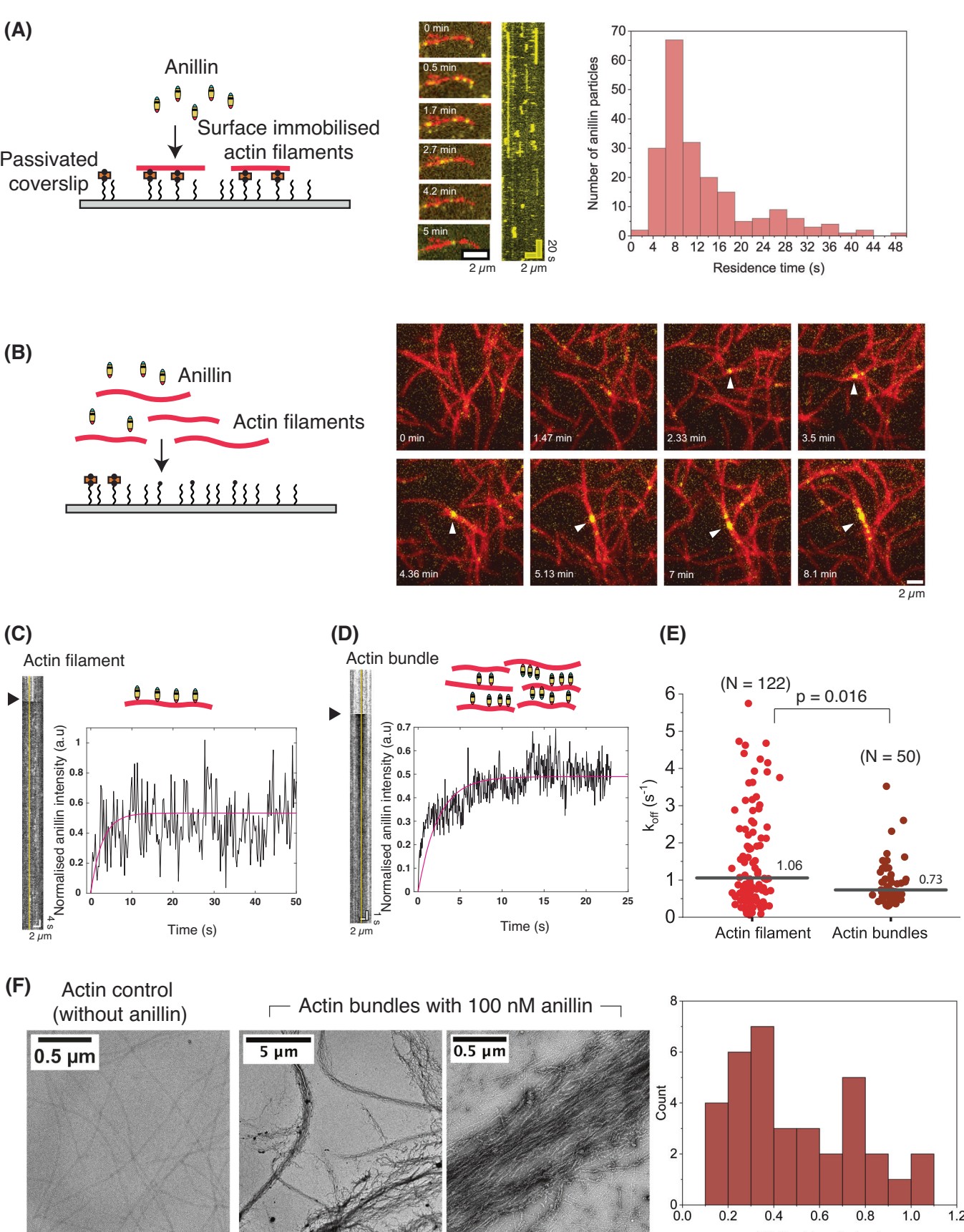

◀ **Figure 3.  Anillin binds F-actin and forms heterogenous and loose F-actin bundles.**

(A) Left: Time-lapse TIRF image series and corresponding kymograph of 10 nM anillin (yellow), showing binding and unbinding from a surface-bound actin filament (red). Right: Distribution of residence times of anillin particles on surface-bound actin filaments ($N = 203$). (B) Time-lapse image series showing an F-actin bundle being formed by 5 nM anillin. With time, more actin filaments (red) are recruited, and correspondingly, the anillin intensity (yellow) at the filament junctions also increases. White arrow points to the formation of the bundle. (C, D) Kymographs and corresponding recovery curves after photobleaching of anillin on an actin filament and actin bundle, respectively (see schematics). A line is drawn (5-pixel wide) on the kymograph (shown in yellow) to obtain the intensity profiles. Normalisation was done by first subtracting the fluorescence intensities from the minimum intensity after bleaching and then dividing by the maximum intensity before bleaching, which was taken as the average of the values pre-FRAP. Pink solid curves show single exponential fits. Respective $R^2$ values for (C, D): 0.06658, 0.5141. (E) Off-rates ($k_{off}$) obtained from single exponential fits of recovery curves for anillin bound to different components. The horizontal lines represent the median values. $P = 0.016$ using a Mann–Whitney $U$ test. (F) Electron micrographs of actin filaments in the absence of anillin (left; re-displayed in Fig. EV2C for reference) and actin bundles formed in the presence of 100 nM anillin at low and high magnification, respectively (right). The histogram shows the distribution of F-actin bundle widths from 30 bundles in 15 electron micrographs. For bundles of variable width, the bundle width was determined across different regions of the bundle. Source data are available online for this figure.

dynamically along a single microtubule. Occasionally, we observed instances where multiple actin filaments bound to overlapping positions on a microtubule (Fig. 4A, Movie EV3). This suggests either binding to distinct protofilaments within the microtubule or the stacking of one actin filament atop another already bound to the microtubule. Interestingly, in cases where a single actin filament partially overlapped with a microtubule, we observed directional sliding of the filament along the microtubule until the overlap length was maximised (Fig. 4B). In regions of overlap between actin filaments and microtubules, the anillin signal increased upon the recruitment of actin filaments to the microtubule surface (Fig. 4C(i)). Furthermore, when the actin filament was shorter than the microtubule seed, the entire microtubule-bound anillin signal redistributed to the overlap (Fig. 4C(i)). At 10 nM anillin, we observed a gradual increase of the intensity of anillin in these overlaps as more actin filaments got recruited (Fig. 4C(ii),(iii)). At 100 nM anillin, the increase in anillin signal was simultaneous to the recruitment of actin filament(s) (Fig. EV4A). These observations indicate a strong preference of anillin for actin–microtubule overlaps. This results in relative sliding of overlapping filaments in the direction increasing the overlap length.

To visualise anillin-mediated actin–microtubule interactions at a higher spatial resolution, we used negative-stain EM. When GMPCPP microtubules, F-actin and anillin (100 nM) were mixed together in solution before deposition on the EM grid, we could see large complexes containing microtubule bundles with actin filaments in between (Fig. 4D, right). Such microtubule bundles were not observed when only microtubules and anillin (Fig. 4D, left) or only actin and microtubules (Fig. 4D, left) were present.

We also performed the FRAP assay to obtain $k_{off}$ values for anillin bound to actin–microtubule overlaps (Fig. 4E). The median off-rate for anillin at actin–microtubule overlaps was comparable to the off-rate measured at GMPCPP microtubules and microtubule-microtubule and actin–actin overlaps (Fig. 4F). The off-rate was consistently lower than the off-rates measured at single actin filaments and GDP microtubules, showing a preference of anillin for the actin–microtubule overlaps.

To test whether anillin could likewise crosslink actin filaments to dynamic microtubules, we grew microtubules from GMPCPP seeds for ~5 min before adding a solution with tubulin dimers, anillin, and pre-polymerised, phalloidin-stabilised actin filaments. This procedure was designed to prevent anillin from encountering only the microtubule seeds first and accumulating there. We used 10 nM anillin to avoid premature bundling of actin filaments before reaching the microtubules. As shown in Fig. 5, we observed anillin-

mediated recruitment of actin filaments onto dynamic microtubules, confirming that anillin can also crosslink actin filaments to dynamic microtubules. F-actin recruitment occurred at different locations on the microtubule lattice and various forms of interactions could be observed ($n = 66$ total events of actin filament(s) recruited to a microtubule). Some actin filaments bound and unbound (Fig. 5A; Movie EV4). In other cases, we observed actin filaments sliding on the microtubule lattice, accompanied by diffusion of anillin molecules (Fig. 5B; Movie EV5). A subset of these sliding events involved tip tracking of actin filaments on growing microtubules, facilitated by anillin (Fig. 5C, observed in 18.2% of cases; Movie EV6). Finally, in some cases, multiple actin filaments were recruited (Fig. 5D; Movie EV7). It was observed that when multiple actin filaments were recruited, the residence time on the microtubule was increased (mean value of 6.7 s, $n = 28$) as compared to when a single filament was recruited (mean residence time on the microtubule of 2 s, $n = 32$), as shown in Fig. 5E. In all cases, regions of overlap between actin and microtubules had a higher intensity of anillin as compared to other regions. The distribution of the location of actin recruitment on the dynamic microtubules is summarised in Fig. 5F.

# Discussion

In this paper, we demonstrated that anillin functions as a crosslinking protein, directly bridging microtubules and actin filaments. This finding aligns with the broader understanding that the microtubule and actin cytoskeletons are not isolated networks but instead interact through a variety of mechanisms, including direct protein-mediated crosslinking, signalling loops, and transcriptional regulation (Dogterom and Koenderink, 2019; Coles and Bradke, 2015; Pimm and Henty-Ridilla, 2021). Both cytoskeletal systems play critical roles in processes such as cell division, motility, organelle transport, and tissue morphogenesis, and their organisation and dynamics are tightly regulated by accessory proteins that mediate these interactions. Anillin's newly identified role as a crosslinking protein provides additional insight into how such coordination is achieved at the molecular level.

Anillin, known primarily as a central scaffold protein in dividing cells and at cell–cell junctions, has been studied extensively in the context of its actin-bundling function (Jananji et al, 2017; Matsuda et al, 2020; Kučera et al, 2021). Here, using an in vitro reconstitution approach, we systematically investigated the interactions of anillin with microtubules and F-actin separately and in

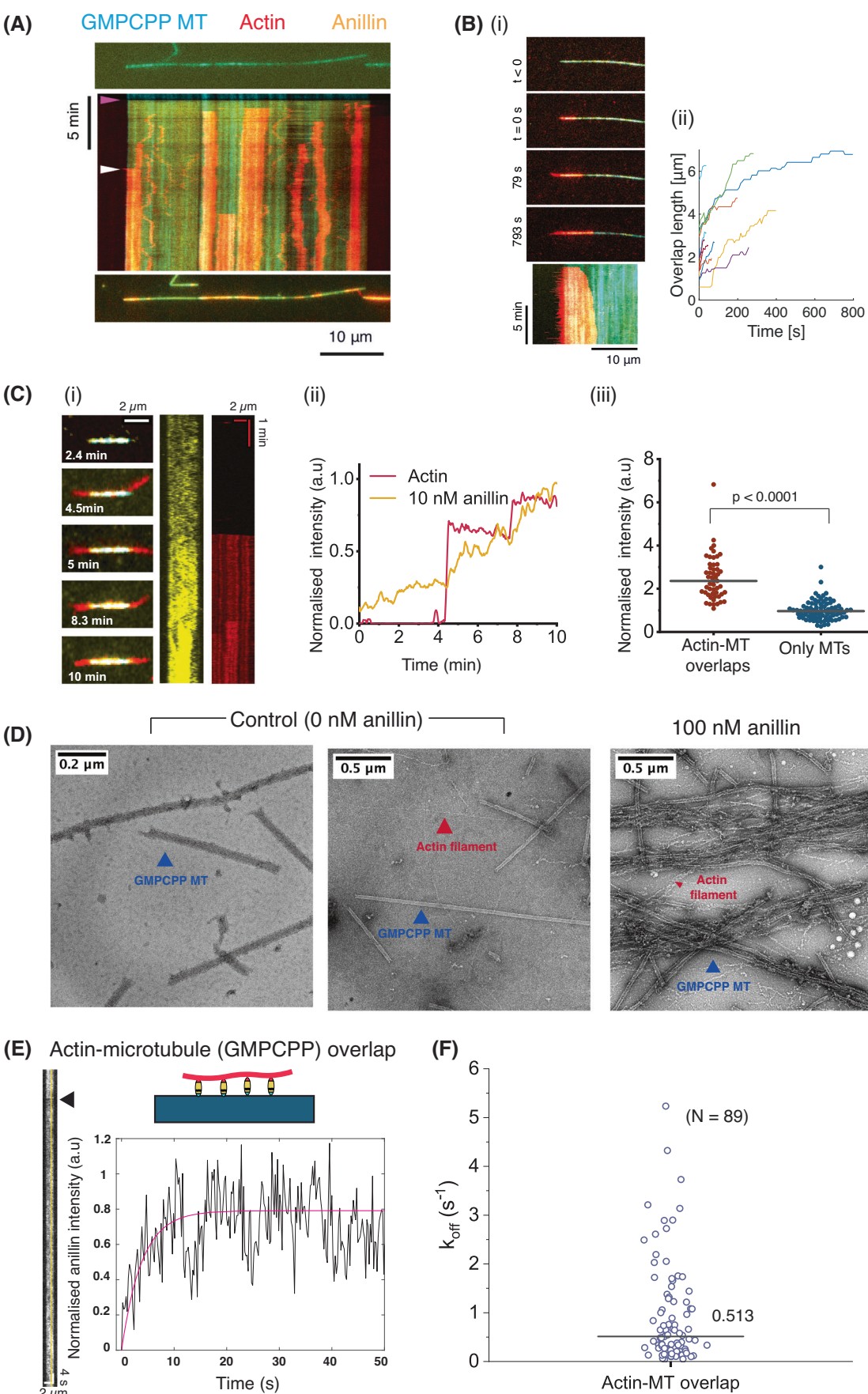

**Figure 4. Anillin crosslinks actin filaments to stabilised microtubules.**

(A) Kymograph showing short actin filaments (red) getting recruited to a long GMPCPP microtubule (cyan, see still image on top) by anillin (yellow) and diffusing and sliding over it. Purple arrowhead indicates the time point where 12 nM anillin was flown in, white arrowhead indicates the sliding event. (B) (i) Time-lapse image series (top panels) and kymograph (bottom panel) showing directional sliding of an actin filament on a GMPCPP microtubule, leading to increase of the overlap length. (ii) Quantification of the increase in actin–microtubule overlap length with time in the presence of anillin for different actin filaments. (C) (i) Time-lapse image series (left) and kymographs (middle, right) showing the increase in anillin (yellow) intensity when subsequent actin filaments (red) are captured on a GMPCPP microtubule (cyan). (ii) Normalised intensities of actin and anillin from the data in (i). The fluorescence values of actin or anillin after background subtraction were divided by their maximum values. (iii) Comparison of mean normalised fluorescence intensities of anillin on actin–microtubule overlaps (measured in the absence of methylcellulose, to avoid the capture of multiple actin filaments on microtubules; $N = 51$ overlaps) vs GMPCPP microtubules ($N = 82$ filaments). In both cases, the intensities were normalised by background subtraction and divided by the mean value for anillin only on microtubules. Exact $P = 6.09 \times 10^{-24}$ using a Mann–Whitney $U$ test. (D) Electron micrograph of actin filaments (red arrowhead) and GMPCPP microtubules (blue arrowhead) in the absence (left) and presence (right) of 100 nM anillin. (E) Kymograph and corresponding recovery curve after photobleaching of anillin on an actin–microtubule (GMPCPP) overlap. A line is drawn (5-pixel wide) on the kymograph (shown in yellow) to obtain the intensity profile. Normalisation was done by first subtracting the fluorescence intensities from the minimum intensity after bleaching and then dividing by the maximum intensity before bleaching, which was taken as the average of the values pre-FRAP. Pink solid curve shows the single exponential fit ($R^2$ value = 0.164). (F) Off-rate ($k_{off}$) obtained from single exponential fits of recovery curves for anillin bound to actin–microtubule overlaps. The horizontal line represents the median value. Source data are available online for this figure.

combination (see summary in Fig. 6), showing that anillin can directly bind to microtubules as well as actin filaments and that anillin can mediate crosslinking between microtubules and actin filaments.

For anillin interacting with actin filaments, we observed that anillin prefers to bind actin bundles over single actin filaments, consistent with prior reports using *Drosophila* anillin (Field and Alberts, 1995; Jananji et al, 2017) and human anillin (Matsuda et al, 2020; Kučera et al, 2021). In our assays, anillin tends to form actin bundles that are loose and heterogeneous in width, unlike other more well-studied actin-bundling proteins such as fascin or α-actinin (Winkelman et al, 2016), or even anillin itself in other reports (Kučera et al, 2021). This difference is likely attributable to the distinct ionic composition of the buffer used here, designed to support dynamic microtubules, which may have altered anillin's affinity for actin filaments.

While co-sedimentation assays have previously implicated the association of anillin with microtubules (Tse et al, 2011; van Oostende Triplet et al, 2014; Sisson et al, 2000), and the potential microtubule-binding region of anillin has been refined to an area located in the C-terminal half of the protein (van Oostende Triplet et al, 2014), little was known about the nature of this association. Here, we have shown anillin's preferential binding to GMPCPP, GTP and taxol-stabilised microtubules over a GDP lattice. This ability of microtubule-associated proteins to recognise lattice types is relatively common and is likely influenced by differences in lattice spacing (preprint: Liu et al, 2024; Siahaan et al, 2022). It may serve a biological function, as it has been proposed to underlie the mechanism of tip tracking (Reid et al, 2019) and potentially play a regulatory role (Siahaan et al, 2022). This fact may explain why, in our experiments, anillin not only binds the lattice of dynamic microtubules but also tracks growing and shrinking plus ends. Indeed, anillin does not have any known tip localisation signal, and unlike other plus-end tracking actin–microtubule crosslinkers such as Gas2L1 (van de Willige et al, 2019) and MACF/ACF7 (Wu et al, 2008), it tracks the tip autonomously without requiring EB1. Whether the lattice-type recognition can explain the enhanced catastrophe and rescue frequencies of dynamic microtubules that we observed remains to be seen in future research.

Our experiments also revealed the co-existence of multiple sizes of anillin particles diffusing on the microtubule lattice. One possible source for clustering or, potentially, multimerization is

the putative coiled-coil domain predicted based on the anillin sequence (Piekny and Maddox, 2010). The tendency of anillin to form larger assemblies aligns with prior observations. Its yeast homologue, Mid1p, has been shown to undergo liquid–liquid phase separation driven by a long, unstructured N-terminal domain (Chatterjee and Pollard, 2019). PONDR-FIT analysis in the same study predicted extensive disordered regions at the N-terminus of human anillin, suggesting a comparable mechanism. AlphaFold predictions for isoform 2 used in this work support this possibility (Appendix Fig. S3; also available for isoform 1 (1124 a.a, UniProt ID Q9NQW6, AlphaFold identifier AF-Q9NQW6-F1)). Microtubule binding may potentially promote anillin condensation or clustering by increasing the local anillin concentration, possibly accompanied by conformational changes in anillin and the adoption of a more ordered stable conformation. The clustering of anillin may also explain why anillin can form microtubule bundles, given that so far, only one microtubule-binding site has been predicted (van Oostende Triplet et al, 2014). This is unlike crosslinkers such as PRC1 which in vitro was shown to bundle microtubules by being present as a pre-assembled homodimer (Subramanian et al, 2010). In *C. elegans*, anillin has also been shown to form linear structures that can promote furrow ingression when unbranched F-actin polymerisation is compromised (Lebedev et al, 2023).

We find that anillin molecules bound to microtubules retain their ability to interact with actin filaments, enabling the formation of actin–microtubule bundles. This is similar to tau, which not only bundles microtubules, but can also crosslink microtubules and actin filaments, whereby in vitro it was shown to co-organise and couple the growth of the two networks (Elie et al, 2015). Structurally, the actin-binding and predicted microtubule-binding domains of anillin are located at opposite ends of its sequence, suggesting that anillin can, in principle, form these bundles as monomers. However, as a significant portion of the anillin sequence is intrinsically disordered, these regions may not form well-defined domains in the classical sense, which could influence their interactions. Furthermore, the clustering of anillin on the microtubule lattice likely introduces additional complexity within actin–microtubule overlap regions.

Finally, we find that anillin diffusion drives directional actin filament sliding over microtubules as well as transport by the microtubule tips. Symmetry-breaking, directional sliding, propelled

**(A)** Binding and unbinding

**(C)** Tip tracking

**(B)** Lattice sliding

**(D)** Bundling

Seed+dynamic microtubule    Anillin    Actin

**(E)**    Time spent by actin filaments on microtubules

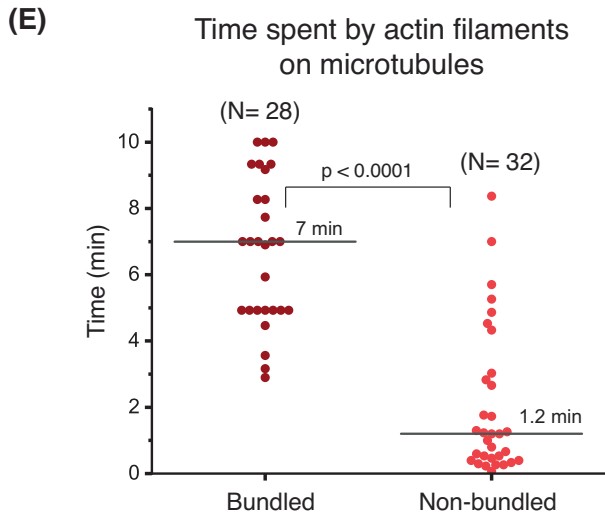

**(F)** Distribution of location of anillin mediated recruitment of actin filaments on microtubules

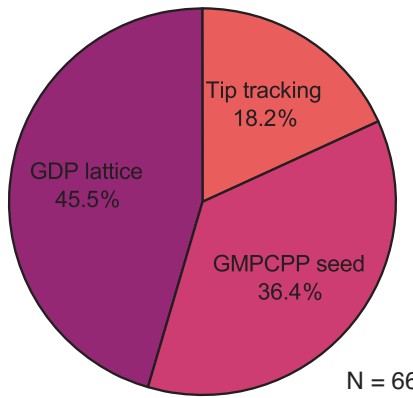

**Figure 5. Anillin-mediated dynamics of actin filaments on dynamic microtubules.**

Schematics and corresponding time-lapse image series and kymographs (actin and anillin channels) of different events that occur upon anillin-mediated capture of actin filaments on dynamic microtubules (light cyan) growing from seeds (dark cyan). Purple arrowhead indicates the + end of the microtubule. (A) An example where actin filaments bind and unbind from a microtubule. The white arrowheads point to the locations where the actin filaments were captured and then unbound. At 4.2 min, one actin filament was captured, which unbound at 4.6 min. At 5.5 min, another actin filament was captured, which unbound at 5.8 min. In the kymographs, white arrows point to the binding and unbinding events shown in the time-lapse. (B) An example where an actin filament slides on the microtubule. The white arrowheads point to the change in location of the actin filament on the microtubule because of sliding. (C) Example of an actin filament that tracks the microtubule end due to anillin. The white arrowhead points to the actin filament being captured on the microtubule + end by anillin and then moving along with the growing and shrinking end. (D) An example where multiple actin filaments get captured on a microtubule. A first filament is captured at 6.7 min. The white arrowhead at 7.2 min points to the location where a second filament gets captured. Note that in this example eventually the bundle also moves with the shrinking microtubule. (E) Distribution of time spent by bundled versus non-bundled actin filaments on microtubules, mediated by anillin. Exact $P = 5.13 \times 10^{-10}$ using a Mann–Whitney $U$ test. (F) Distribution of anillin-mediated recruitment of actin filaments on different locations on a dynamic microtubule. Source data are available online for this figure.

by diffusive filament-crosslinking proteins, has also been observed for other cytoskeletal crosslinkers (Lansky et al, 2015; Alkemade et al, 2022).

Our finding that anillin can serve as a crosslinker between F-actin and microtubules lends strong support to the long-standing idea that anillin contributes to coordination between the microtubule and actin cytoskeleton in dividing cells. Actin–microtubule crosstalk has been implicated in the correct positioning (Yüce et al, 2005; Canman et al, 2008; Foe and von Dassow, 2008) and stabilisation (Bellingham-Johnstun et al, 2023) of the actin cytokinetic ring, and anillin was speculated to play an indirect role in this process by interacting with the microtubule-bound RacGAP (Gregory et al, 2008). Our data prove that anillin can directly crosslink actin filaments and microtubules without accessory proteins. We showed that one consequence of this crosslinking action is the transport of actin filaments by growing and shrinking microtubule tips. This effect could perhaps contribute to the ability of equatorial microtubules to position the contractile actin-myosin ring in the middle of the cell. The strong effect on microtubule rescues and related enhancement of microtubule stability by anillin could further help in these processes. It was previously shown that taxol-stabilised microtubules can correctly position the cytokinetic furrow in mammalian cells (Shannon et al, 2005). Our observations that anillin has a preference for a GTP-like microtubule lattice could therefore have direct implications for the midpoint localisation during cell division. In addition, it is important for proper cytokinesis that contractile proteins, including anillin, are localised primarily at mid-cell rather than around the poles in order to promote furrow ingression. It was previously hypothesized that astral microtubules could help in removing anillin from polar cortices, ensuring anillin is mostly present at the equatorial cortex for faithful furrow ingression (Akhshi et al, 2014). Our results suggest that astral microtubules could indeed directly interact with anillin at the polar cortices. Our in vitro observations indicate that anillin has a higher binding affinity and slower off-rate from overlaps of actin filament and microtubules as compared to single filaments, promoting its function as a crosslinker.

Understanding the regulation of anillin localisation is key to elucidating its role in cytoskeletal crosstalk. During interphase, anillin is predominantly localised in the nucleus, but as the cell progresses through the cycle, it transitions to the cytoplasm, where it interacts with the cytoskeleton (Piekny and Maddox, 2010). This cell cycle-dependent redistribution suggests that anillin's localisation and function may be regulated by simultaneous interactions with various partners, such as motor proteins (e.g., myosin II),

septins, lipids, and signalling proteins like RhoA and Ect2. Furthermore, post-translational modifications of anillin (Kim et al, 2017) and its potentially variable affinities for different isoforms of its binding partners (Chen et al, 2021) are likely to contribute significantly to its regulation, although these mechanisms have yet to be fully characterised.

In this study, we provided new insights into anillin's role in cytoskeletal organisation by demonstrating its ability to crosslink microtubules and actin filaments directly and move them relative to one another. These findings add a novel dimension to anillin's established association with the cytoskeleton, revealing its capacity to physically integrate the two cytoskeletal systems. Such crosslinking activity is likely to be particularly relevant during cell division, where coordination between microtubules and actin filaments is essential (Straight and Field, 2000; Advedissian et al, 2024) and where many anillin-dependent processes are already known to occur (Kechad et al, 2012; Zanin et al, 2013). Beyond cytokinesis, anillin's crosslinking function may also contribute to neural development and migration, where actin and microtubule dynamics are tightly coordinated and interdependent (Leterrier et al, 2017). Furthermore, understanding how these functions are implicated in pathological conditions, such as the dysregulation of anillin expression observed in cancers (Cui et al, 2022), could provide valuable insights into mechanisms underlying disease progression.

## Methods

**Reagents and tools table**

| Reagent/resource | Reference or source | Identifier or catalogue number |
|---|---|---|
| **Antibodies** | | |
| Monoclonal Anti-β-Tubulin I antibody produced in mouse | Sigma-Aldrich | T7816 |
| **Chemicals, enzymes and other reagents** | | |
| Tubulin Protein (>99% Pure): Porcine Brain | Cytoskeleton, Inc. | T240-C |
| Tubulin Protein (Rhodamine): Porcine Brain | Cytoskeleton, Inc. | TL590M-B |
| Tubulin Protein (Fluorescent HiLyte 647): Porcine Brain | Cytoskeleton, Inc. | TL670M-B |
| Tubulin Protein (Biotin): Porcine Brain | Cytoskeleton, Inc. | T333P-B |

| Reagent/resource | Reference or source | Identifier or catalogue number |
|---|---|---|
| GMPCPP (Guanosine-5'-[(α,β)-methylene] triphosphate, Sodium salt | Jena Biosciences | NU-405L |
| PLL-PEG-biotin | SUSOS | (PLL(20)-g[3.5]-PEG(2)/PEG(3.4)-Biotin(20%) |
| Neutravidin | Thermo Fisher Scientific | 31050 |
| κ-casein | Sigma-Aldrich | C0406 |
| PIPES (piperazine-N,N'-bis(2-ethanesulfonic acid) | Sigma-Aldrich | P6757 |
| EGTA (ethylene glycol-bis(β-aminoethyl ether)-N,N,N',N'-tetraacetic acid) | Sigma-Aldrich | E3889 |
| Magnesium Chloride | Sigma-Aldrich | M8266 |
| Pluronic F127 | Sigma-Aldrich | P2443 |
| Methylcellulose | Sigma-Aldrich | M0512 |
| DTT (DL-Dithiothreitol) | Sigma-Aldrich | D0632 |
| Catalase | Sigma-Aldrich | C1345 |
| Glucose Oxidase | Sigma-Aldrich | G6125 |
| Glucose | Sigma-Aldrich | 68270 |
| GTP (Guanosine-5'-triphosphate sodium salt hydrate) | Sigma-Aldrich | G8877 |
| Glycerol | Sigma-Aldrich | G7893 |
| Taxol | Cytoskeleton, Inc. | TXD01 |
| Actin (rabbit skeletal muscle alpha actin) | Hypermol | 8101-03 |
| Potassium chloride | Sigma-Aldrich | P9333 |
| ATP (Adenosine 5'-triphosphate magnesium salt) | Sigma-Aldrich | A9187 |
| Phalloidin | Sigma-Aldrich | P2141 |
| Phalloidin 568 | Thermo Fisher Scientific | A12380 |
| ATTO550-Actin for TIRFM (alpha-Actin, skeletal muscle rabbit) | Hypermol | 8156-01 |
| ATTO647-Actin for TIRFM (alpha-Actin, skeletal muscle rabbit) | Hypermol | 8158-01 |
| Biotin-actin | Hypermol | 8109-01 |
| Uranyl acetate | Electron Microscopy Sciences | 541-09-3 |
| Isopropanol | Honeywell | 33539 |
| Hydrogen peroxide | Sigma-Aldrich | 31642 |
| Ammonium hydroxide | Sigma-Aldrich | 221228 |
| Protease inhibitor cocktail | Roche Diagnostics GmBH | 34044100 |
| Benzonase (0.63 U/mL) | Novagen | 70664 |
| **Software** | | |
| MATLAB | Mathworks Inc. | |

| Reagent/resource | Reference or source | Identifier or catalogue number |
|---|---|---|
| FiJi | Schindelin J et al (2012). https://doi.org/10.1038/nmeth.2019 | |
| Origin (Pro), version 2022 | OriginLab Corporation, Northampton, MA, USA | |
| **Other** | | |
| Cu400 carbon support grids | Quantifoil | N1-C73nCu40-01 |
| Microscope slides | Menzel-Gläser, 76 × 26 mm or Epredia | BC026076A140 |
| Coverslips | Menzel Gläser, 24 × 24 mm, thickness 1 | BB024024A123 |
| Coverslips | Marienfield, 24 × 50 mm, No. 1.5H | 0107222 |
| NiNTA agarose resin | Thermo Fisher Scientific | XF340049 |

## Methods and protocol

### Expression and purification of Anillin-GFP

Anillin-GFP was expressed and purified as described previously (Kučera et al, 2021). Homo sapiens anilin cDNA (GeneBank accession number: BC070066) was ligated into a pOCC vector backbone containing a C-terminal GFP, a 3 C protease cleavage site, and a 6xHisTag. This construct was expressed in Sf9 cells using the FlexiBAC baculovirus expression system (Lemaitre et al, 2019). Sf9 cells were infected by baculovirus stock, shaken moderately (120 RPM) at 27 °C, and harvested 3 days after infection (centrifugation at $300 \times g$, 10 min). The cell pellet was then resuspended in 5 ml of PBS, snap-frozen in liquid nitrogen, and stored at −80 °C for further use. For purification, cell pellet in PBS was quickly thawed and resuspended in lysis buffer (50 mM Na–phosphate buffer, pH 7.5, 30 mM Imidazole, 5% glycerol, 300 mM KCl, 1 mM $MgCl_2$, 0.1% Tween-20, 10 mM β-mercaptoethanol, 0.1 mM ATP, benzonase (0.63 U/mL, 70664, Novagen), 1× Protease inhibitor cocktail (34044100, Roche Diagnostics GmBH), followed by centrifugation at $70,000 \times g$ (1 h, 4 °C) to lyse the cells and collect the supernatant containing Anillin-GFP protein. The supernatant was then incubated with lysis buffer equilibrated NiNTA agarose resin (HisTrap, XF340049, Thermo Scientific) at 4 °C for 2 h while slowly rotating. Resin was then introduced into the column and washed extensively with wash buffer (50 mM Na–phosphate buffer, pH 7.5, 50 mM Imidazole, 5% glycerol, 300 mM KCl, 1 mM $MgCl_2$, 0.1% Tween-20, 10 mM β-mercaptoethanol, 0.1 mM ATP). To remove the 6xHisTag, the resin-bound protein was incubated with PreScission protease (3C HRV protease, 1:100 (1 μg enzyme/100 μg of protein)) overnight, at 4 °C, while rotating. Anillin-GFP released from the resin by PreScission cleavage was collected and concentrated using a 100 kDa centrifugal filter tube (Amicon Ultra-15, Merck). The purity of the protein was evaluated using SDS-PAGE (Appendix Fig. S1E), and the concentration was estimated from the absorbance at both 280 nm and 488 nm using NanoDrop ND-1000 spectrophotometer (Thermo Scientific). Purified protein was aliquoted, snap-frozen in liquid nitrogen, and stored at −80 °C.

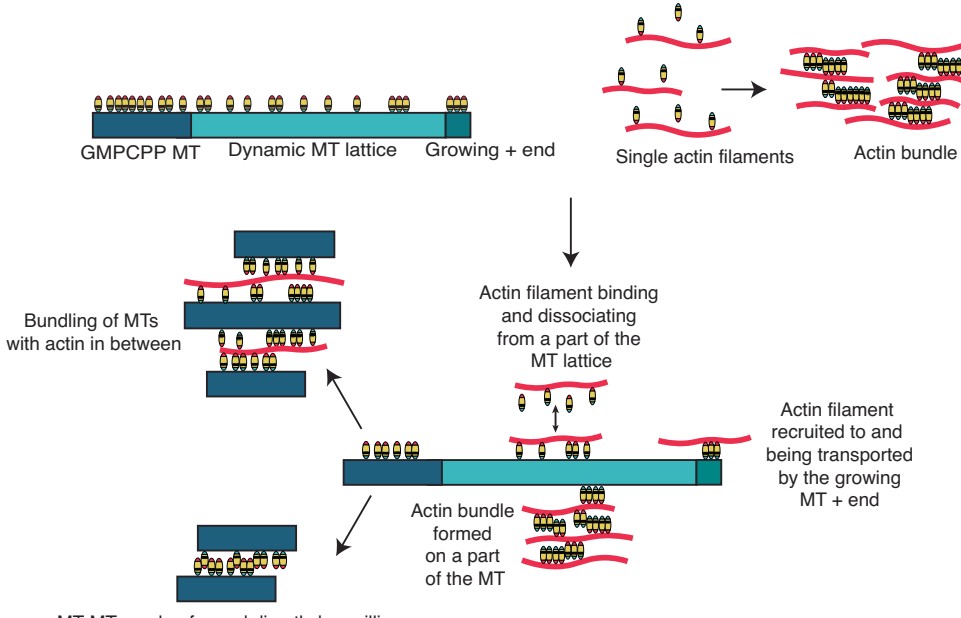

**Figure 6. Schematic summary of anillin bound to a microtubule, actin (single filaments or bundles) and actin–microtubule overlaps.**

On the microtubule, anillin can be present as monomers, dimers or even larger oligomeric species. On single actin filaments, anillin binds and unbinds as monomers, whereas on bundles it can dimerise (and potentially multimerise). Anillin can crosslink actin filaments to microtubules. Crosslinking can cause the capture of actin filaments on surface-immobilised microtubules at different locations (seed, GDP lattice or the + end tip), and with different consequences (binding and unbinding of actin filaments, sliding of actin filaments, bundling of actin filaments, or tip tracking of actin filaments). In actin–microtubule overlaps, anillin is present as a mixture of monomers, dimers, and potentially even larger oligomeric species. Lastly, microtubules can also be bundled by either anillin on its own, or by anillin and actin together.

### Preparation of GMPCPP stabilised, doubly cycled microtubule seeds

Biotinylated and fluorescently labelled microtubule seeds were prepared from a solution containing a dimer concentration of 8.34 µM porcine tubulin (a mixture of 5.6% biotinylated tubulin, 30% tubulin labelled with either Rhodamine or HiLyte 647 and 64.4% unlabelled tubulin; all from Cytoskeleton inc. via Tebu-Bio) and 1 mM GMPCPP (Guanosine-5'-[(α,β)-methylene] triphosphate, Sodium salt, 10 mM stock, Jena Biosciences) in MRB80 buffer. For non-biotinylated seeds, a tubulin mixture of 30% fluorescently labelled tubulin and 70% unlabelled tubulin was used. After incubating for 5 min on ice, tubes containing the tubulin solutions were transferred to a 37 °C water bath and incubated for 30 min to promote microtubule polymerisation. The solution was then airfuged using a Beckman Coulter airfuge with the rotor prewarmed to 37 °C at 30 psi for 5 min. The supernatant was discarded and the microtubule pellet was resuspended in 18 µL MRB80 (final concentration 2.67 µM, assuming 80% recovery). The resuspended solution was incubated on ice for 20 min to depolymerise the microtubules. 1 mM GMPCPP was then added, followed by another 5 min incubation on ice. The solution was again transferred to the 37 °C water bath, incubated for 30 min, and airfuged with the rotor prewarmed to 37 °C at 30 psi for 5 min. The supernatant was discarded and the microtubule pellet was resuspended in 20 µL warm MRB80 containing 10% glycerol. In total, 2 µL aliquots were snap-frozen using liquid nitrogen and stored at −80 °C for later use.

### Channel preparation and surface treatment for TIRF assays

Microscope slides (Menzel-Gläser, 76 × 26 mm or Epredia) and coverslips (Menzel Gläser, 24 × 24 mm, thickness 1) were cleaned using Base Piranha (Preciado López et al, 2014a). After 3× washes with MillliQ, these were dried using nitrogen. Strips were cut from parafilm as spacers, which were then placed between a slide and coverglass. The chambers were then placed briefly on a hot surface to form a seal between the slide and coverglass. Before experiments, the channels were passivated by injecting 0.2 mg/mL of PLL-PEG-biotin (PLL(20)-g[3.5]-PEG(2)/PEG(3.4)-Biotin(20%), SUSOS, 2 mg/mL stock in MRB80 buffer (80 mM PIPES (piperazine-N,N'-bis(2-ethanesulfonic acid)), 1 mm EGTA (ethylene glycol-bis(β-aminoethyl ether)-N,N,N',N'-tetraacetic acid), 4 mM MgCl$_2$, pH 6.8)) into the channels and incubating for 10 min. After 4× washes with MRB80, the surface was coated by the addition of 0.2 mg/mL neutravidin (Sigma, 5 mg/mL stock in MRB80) and incubated for 10 min. After 4× MRB80 washes, finally, 0.5 mg/mL κ-casein (Sigma, 5 mg/mL stock in MRB80) was added and incubated for 10 min, followed by 4× MRB80 washes. Alternatively, the channels were functionalised by tubulin antibodies (Sigma-Aldrich, T7816, 10 µg/ml) for 10 min, followed by passivation with 1% Pluronic F127 (P2443, Sigma) for 1 h.

### TIRF assay for probing anillin binding to dynamic MTs

Biotinylated microtubule seeds were diluted in the ratio 1:200 (v/v) in warm MRB80, flown into the channels and incubated for 10 min. The channels were then washed 4–5× with MRB80 to remove unattached seeds. A tubulin mix containing 0.5 mg/mL κ-casein, 0.1% methylcellulose (w/v in MRB80), oxygen scavenger (4 mm DTT, 0.2 mg/mL catalase, 0.4 mg/mL glucose oxidase), 50 mM glucose, 1 mM GTP, 15 µM tubulin (of which 14.5 µM unlabelled and 0.5 µM labelled with either Rhodamine or Hilyte 647) was

prepared on ice and airfuged at 30 psi for 5 min (rotor at 4 °C) to remove any aggregates. When using anillin, the desired concentration of the protein was added to the supernatant. The supernatant was then injected into the channel. In this and all other TIRF assays, the channel was sealed on both ends with VALAP (1:1:1 vaseline, lanolin, paraffin) before imaging.

### TIRF assay for probing anillin binding to taxol-stabilised microtubules

A solution containing 70 µM tubulin (2% biotinylated, 2% fluorescently labelled with either HiLyte 647 or Rhodamine), 1 mM GTP and 27% glycerol was prepared on ice and then transferred to a 37 °C water bath for 20 min. 25 µM Taxol (Cytoskeleton Inc., dissolved in DMSO) was added and the solution further incubated for 10 min. This solution was then airfuged for 3 min at 14 psi (rotor at 37 °C). The pellet was washed with MRB80 containing 40 µM taxol and resuspended in a fresh 40 µM taxol solution. This solution was diluted according to the filament distribution needed for imaging, and anillin was added before injecting into the channels. The channels were sealed on both ends using wax before imaging.

### TIRF assay for probing anillin binding to F-actin

Actin filaments were polymerised in MRB80 containing 1 µM G-actin (rabbit skeletal muscle unlabelled, Hypermol), 50 mM KCl, 1 mM DTT and 0.5 mM Mg ATP. Filaments were allowed to polymerise at room temperature for 30 min followed by the addition of 1 µM Phalloidin 568 (Sigma-Aldrich), corresponding to a 1:1 molar ratio to G-actin. Alternatively, fluorescently labelled G-actin from Hypermol (Atto 550 or Atto 647) was mixed with unlabelled G-actin (10% labelling, total concentration was again 1 µM) and polymerised, then stabilised using unlabelled phalloidin. When biotinylated filaments were needed, biotin-actin was used in a 1:20 molar ratio with remainder actin.

To probe anillin binding to surface-immobilised F-actin, channel preparation was done as described for the dynamic microtubule assay, using biotin, neutravidin and κ-casein. Biotinylated actin filaments were diluted to a final concentration of 200 nM actin (G-actin) in MRB80 and next flown through the channel and incubated for ~5 min, followed by 3× washes with MRB80 to remove untethered filaments. Solutions containing either 10 or 100 nM anillin were prepared in MRB80 with 0.5 mg/mL κ-casein, oxygen scavenger and 50 mM glucose and then flown on top of the actin filaments to check for binding.

To probe anillin binding and bundling activity with freely floating F-actin, we first performed surface passivation of the channels as described above. Non-biotinylated actin filaments were diluted to a final concentration of 100 or 200 nM actin (G-actin) along with 0.5 mg/mL κ-casein, 0.2% methylcellulose, Oxygen scavenger, 50 mM Glucose, 1 mM GTP, 0.5 mM ATP, 1 mM DTT and varying concentrations of anillin in MRB80. The solution was then flown into the channel.

### TIRF assays for probing anillin-mediated actin–microtubule crosstalk

To probe interactions of F-actin with stable GMPCPP microtubule seeds, HiLyte 647-labelled biotinylated microtubule seeds were diluted in a 1:200 (v/v) ratio in warm MRB80 and then flown into the channels. After a 10 min incubation, the channels were washed 4–5 times with MRB80 to remove unattached seeds. A solution containing *imaging buffer* (0.5 mg/mL κ-casein, 0.2%

methylcellulose, 1× Oxygen scavenger, 50 mM glucose, 1 mM GTP, 0.5 mM ATP, 1 mM DTT), 50 nM polymerised F-actin (568 phalloidin -labelled and stabilised), and either 10 or 100 nM anillin was prepared and injected into the channel containing the GMPCPP microtubule seeds.

To probe interactions of F-actin with dynamic microtubules, HiLyte 647-labelled biotinylated microtubule seeds were diluted in the ratio 1:200 (v/v in warm MRB80) and then flown into the channels and incubated for 10 min. The channel was then washed with MRB80 4× or 5× to remove unattached seeds. A tubulin mix containing imaging buffer, 15 µM tubulin (14.5 µM unlabelled and 0.5 µM labelled with Hilyte 647) was prepared on ice and then airfuged at 30 psi for 5 min (rotor at 4 °C). The solution was injected into the channels to grow dynamic MTs for ~5 min at 30 °C, followed by injecting a solution containing the tubulin mix (airfuged), 50 or 100 nM pre-polymerised and stabilised actin (Rhodamine labelled) and 10 nM anillin.

### TIRF assays for probing anillin at MT-MT overlaps

Biotinylated GMPCPP MT seeds labelled with HiLyte 647 tubulin were diluted in warm MRB80 in a 1:100 v/v ratio and then flown into the channels and incubated for 10 min. The channel was then washed with MRB80 4–5 times to remove unattached seeds. Non-biotinylated MT seeds labelled with Rhodamine tubulin were first diluted either 1:50 or 1:100, and then 1 µL of the solution was added to a solution containing imaging buffer and either 10 or 100 nM anillin. This solution was then flown into the channel containing the surface-immobilised biotinylated seeds.

### TIRF microscopy and FRAP measurements

TIRF and FRAP data were acquired on two Nikon Eclipse Ti inverted microscopes, each equipped with a motorised stage, CFI Plan Apochromat λ objective (100XH NA1.45 WD.13), Dual Laser Illuminator iLas[2] for TIRF/FRAP (GATACA systems), and either a 2× EVOLVE 512 Camera (EMCCD 512 × 512 pixels) or a Andor iXon Ultra 888. The microscopes were controlled by Metamorph or NIS-Elements software, respectively.

For TIRF microscopy, videos were acquired at an imaging frequency of either 1 or 2 s per frame and with an exposure time of 500 ms. Videos were usually acquired for 10 min. Assays involving dynamic microtubules were performed at 30 °C using a custom-built objective heater to control the temperature. Dual or triple colour imaging was done sequentially with excitation lasers of wavelengths 488, 561 and 642 nm.

For FRAP, 5-6 circular regions of interest (ROIs) per field of view with a diameter of 15 pixels were chosen. Photobleaching was done using 80% laser power (488 laser) after the first 50–60 frames of acquisition. Videos were acquired at an imaging frequency of 200 ms or 46 ms (stream acquisition) between frames for a total time period of maximum ~2 min. For dynamic MTs, acquisition was done at 26 °C.

## Image analysis for TIRF imaging data

All videos were visualised and processed using FIJI (ImageJ) software (Schindelin et al, 2012).

To quantify the dynamic instability behaviour of MTs, we first generated kymographs along lines with a width of 5 pixels drawn on the filaments to be analysed using the 'KymographBuilder'

plugin. To obtain the microtubule growth and shrink rates, points were selected on the desired slopes to obtain the $x$ (position) and $y$ (time) coordinates. A custom-written MATLAB (Mathworks, Inc.) script was used to obtain the rates using these coordinates. The time-weighted values of these rates were then calculated by first assigning weights to the time values by dividing each time value by the total time, and then multiplying the speeds with the corresponding time weights (Alkemade et al, 2022). These weighted speeds were then added to get a final total weighted speed value. The standard error was calculated according to

$$S.E = \sqrt{\frac{\sum w \times (\Delta x)^2}{(\frac{N-1}{N}) \times \sum w}}, \tag{1}$$

where $w$ is the weight, $\Delta x$ is the speed-average weighted speed, and $N$ is the number of events/slopes.

The microtubule catastrophe frequency was calculated by dividing the total number of catastrophes by the total time that microtubules spent growing. Microtubule rescue frequencies were obtained by dividing the total number of rescues by the total time microtubules spent shrinking. The error was then calculated using the following formula:

$$error = \frac{frequency}{\sqrt{n}}, \tag{2}$$

where 'frequency' denotes either the catastrophe or rescue frequency, and $n$ is the number of catastrophes/rescues.

A two-sample Student's $t$ test (with Welch's correction) was performed to check the statistical significance in the microtubule dynamics between control (0 nM anillin) vs 100 nM anillin-bound microtubules.

### Visualisation and analysis of anillin binding to and bundling actin filaments

The residence times for anillin particles diffusing on single actin filaments were obtained from kymographs as the time span between landing on the filament and unbinding. Only particles that showed diffusion were selected, since a static signal could just mean background noise. An actin-bundling event was characterised by the coming together of actin filaments that also corresponded with the increase in anillin intensity. The lengths of actin bundles from electron micrographs were obtained by drawing a straight line (1-pixel wide) using FIJI across different regions of different bundles (30 bundles in 15 electron micrographs) and measuring the length.

### Analysis of the rate of anillin intensity increase in TIRF images of actin-MT overlaps with GMPCPP microtubules

A 1-pixel wide line was drawn in a region where actin filaments could be observed to be recruited on GMPCPP MT seeds in the presence of anillin. $Z$ axis profiles were then generated to get the information on the change in fluorescence intensity with time for the actin and anillin channels. Using a custom-written MATLAB script, these fluorescence intensity values were normalised for both the actin and anillin channels. For both channels, first background intensity (taken by drawing a 1-pixel wide line close to the overlap) was subtracted from the fluorescence values and then divided by the maximum value in the respective channel. A Mann–Whitney $U$ test

was performed to check the statistical significance between the intensity of anillin on microtubules vs microtubule–actin overlaps.

### Visualisation of actin filaments sliding on dynamic microtubules

The binding of an actin filament to a dynamic microtubule was classified by it being present on the microtubule for at least four frames (8 s) and moving along with the anillin particles on the microtubule. A higher intensity of anillin in this overlap region was further used to confirm the actin–microtubule overlap. For an unbinding event the loss of the signal from the actin channel on the microtubule was checked. For a bundling event, a stable increase in the signal in the actin channel on the overlap was checked. For lattice sliding, the change in position of the start of the actin filament on the microtubule lattice was followed. For tip tracking, the presence of the actin filament on the growing tip of the microtubule (along with a higher anillin signal) was checked.

To obtain the residence times of the actin filaments on dynamic microtubules, the time points (on kymographs) that corresponded to the the landing of the actin filament on the microtubule were subtracted from the time point when the actin filament detached from the microtubule. A Mann–Whitney $U$ test was performed to check the statistical significance between the residence time of bundled vs non-bundled actin filaments on microtubules.

### Image analysis for FRAP measurements

Kymographs were first generated for the desired ROIs that had been bleached. On the kymographs, a line of width 5 pixels was drawn along the time scale (vertical axis) on the region for which the FRAP curve was required. Normalised fluorescence recovery curves were calculated using a custom-written MATLAB script by first subtracting the fluorescence intensities from the minimum intensity after bleaching and then dividing by the maximum intensity before bleaching, which was taken as the average of the values pre-FRAP. The curves starting from the time point of bleaching were fit to a single exponential equation to obtain the $k_{off}$ values of anillin using the 'curve fitting' app on MATLAB:

$$y = A(1 - e^{-t \times k_{off}}), \tag{3}$$

where $y$ = normalised fluorescence intensity, $t$ = time, and A and B are constants.

To compare the asymmetric distributions of $k_{off}$ values that were obtained, the median values were used. A Mann–Whitney $U$ test was performed to check the statistical significance when comparing the $k_{off}$ values.

## Electron microscopy

Non-biotinylated actin filaments were prepared as described above using 7.5 μM G-actin (unlabelled) and diluted to a final concentration of 1 μM G-actin along with 1 mM DTT in MRB80. This sample was used as the control sample for unbundled F-actin filaments. To probe anillin-mediated F-actin bundling, 100 nM anillin was added. GMPCPP microtubule seeds (unlabelled) were prepared as described above. These were diluted to 10% (v/v) in MRB80 and used as a control sample for unbundled MT seeds. To probe interactions of GMPCPP seeds with anillin, 100 nM anillin was added to the mix. To observe microtubule-F-actin crosstalk mediated by anillin, a solution was prepared in MRB80 with

1 µM pre-polymerised actin, GMPCPP microtubules (1.5 µL of prepared seeds diluted in 15 µL of final solution mixture) and 100 nM anillin.

Cu400 carbon support grids (Quantifoil, Cat.# N1-C73nCu40-01) were glow discharged in an oxygen plasma using the GloQube-D instrument (Quorum Technologies Ltd) to enhance sample adsorption. A small (4 µL) droplet of the protein sample was deposited onto the grid and incubated for 1 to 2 min to allow protein adsorption. The samples were washed 3× with MilliQ water to remove any surplus protein and salt, with careful blot-drying after each wash. Finally, the samples were stained for 25 s with a 2% aqueous solution of uranyl acetate (Electron Microscopy Sciences). The samples were then blot-dried to remove any remaining liquid. The samples were imaged on a JEM-1400Plus transmission electron microscope (JEOL) operating at an acceleration voltage of 120 kV. Imaging was carried out using a TemCam-F416 CMOS camera (TVIPS) with a resolution of 4k × 4k pixels.

## Data availability

This study includes no data deposited in external repositories. The data that support the findings of this study are available from the corresponding author upon request.

The source data of this paper are collected in the following database record: biostudies:S-SCDT-10_1038-S44318-025-00492-3.

## Peer review information

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

## Acknowledgements

The authors would like to thank Roman Podhájecký for the assistance with experiments. A portion of the experiments was conducted by Eliška Pilátová as a part of her internship in BIOCEV. We would like to thank Celine Alkemade for helpful discussions regarding experimental design and analysis for actin–microtubule crosslinkers. The authors would like to thank Nikolina Šoštarić and Arjen Jacobi for helpful discussions regarding AlphaFold. This work was supported by the NWO/OCW Gravitation program Building a Synthetic Cell (BaSyC) (grant number 024.003.019) to MD, by the NWO Talent Programme which is financed by the Dutch Research Council (project number VI.C.182.004) to GHK, and by the Czech Science Foundation (grant number 23-07703S) to MB. ZL acknowledges the European Research Council grant ERC-2022-SYG 101071583, institutional support from CAS (RVO: 86652036) and Imaging Methods Core Facility at BIOCEV supported by the MEYS CR [LM2023050, Czech-BioImaging].

## Author contributions

**Ilina Bareja**: Conceptualisation; Data curation; Formal analysis; Investigation; Visualisation; Methodology; Writing—original draft; Writing—review and editing. **Ondřej Kučera**: Conceptualisation; Data curation; Formal analysis; Investigation; Visualisation; Methodology; Writing—review and editing. **Irene Istúriz Petitjean**: Data curation; Formal analysis; Investigation; Visualisation; Methodology. **Beatriz Eugenia Orozco Monroy**: Data curation; Formal analysis; Investigation; Visualisation; Methodology. **Ján Sabó**: Resources; Writing—review and editing. **Marcus Braun**: Conceptualisation; Funding acquisition; Investigation; Methodology. **Zdeněk Lánský**: Conceptualisation; Resources; Funding acquisition; Investigation; Methodology; Writing—review and editing. **Gijsje H Koenderink**: Conceptualisation; Resources; Investigation; Methodology; Writing—review and editing. **Marileen Dogterom**:

Conceptualisation; Resources; Funding acquisition; Investigation; Methodology; Project administration; Writing—review and editing.

Source data underlying figure panels in this paper may have individual authorship assigned. Where available, figure panel/source data authorship is listed in the following database record: biostudies:S-SCDT-10_1038-S44318-025-00492-3.

## Disclosure and competing interests statement

The authors declare no competing interests.

# Expanded View Figures

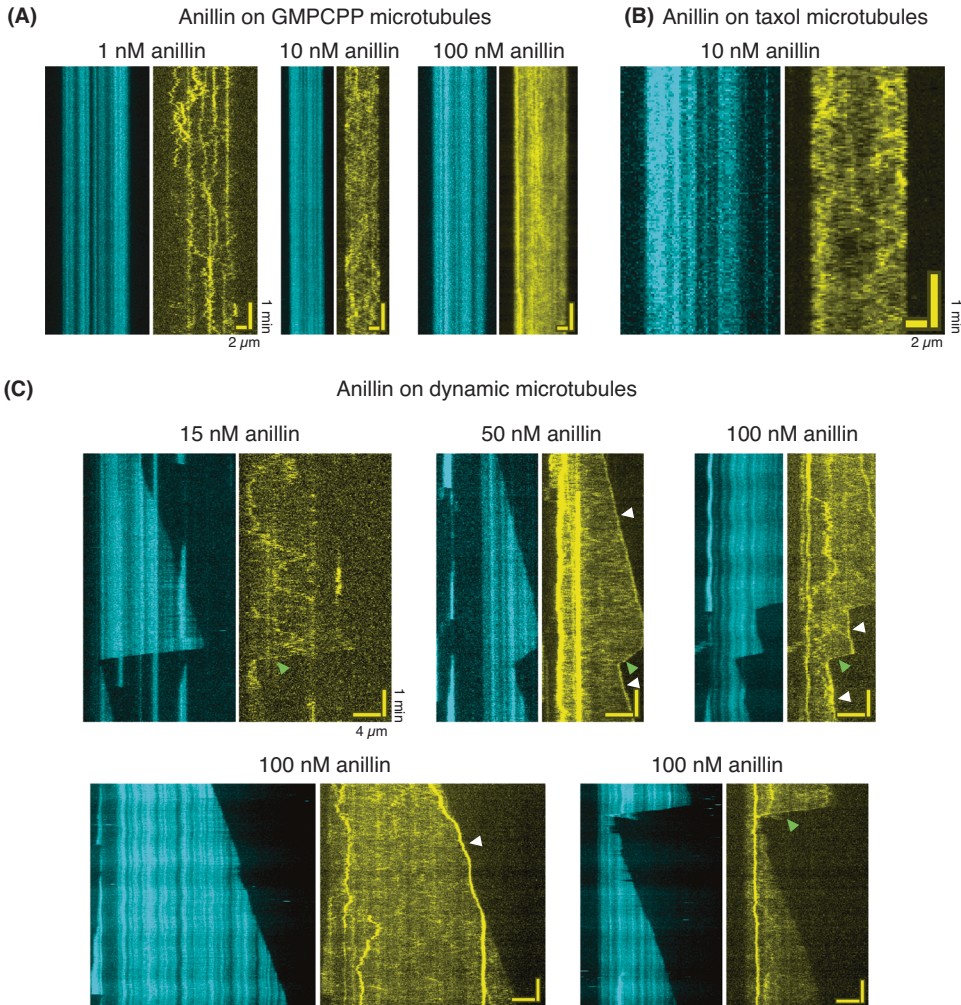

**Figure EV1.  Anillin binds to microtubules.**

(A) Kymographs of anillin-GFP (yellow) diffusing on GMPCPP microtubules (cyan) at different concentrations (see legend). (B) Kymograph of 10 nM anillin-GFP (yellow) diffusing on a taxol microtubule (cyan). (C) Example kymographs of anillin-GFP (yellow) diffusing on dynamic microtubules (cyan) and tracking their + ends at different concentrations (see legend). White arrowheads point to the accumulation of anillin on growing microtubule ends, whereas green arrowheads point to accumulations of anillin on shrinking microtubule ends. Vertical axis denotes time, horizontal axis denotes space (see legends for scale bars).

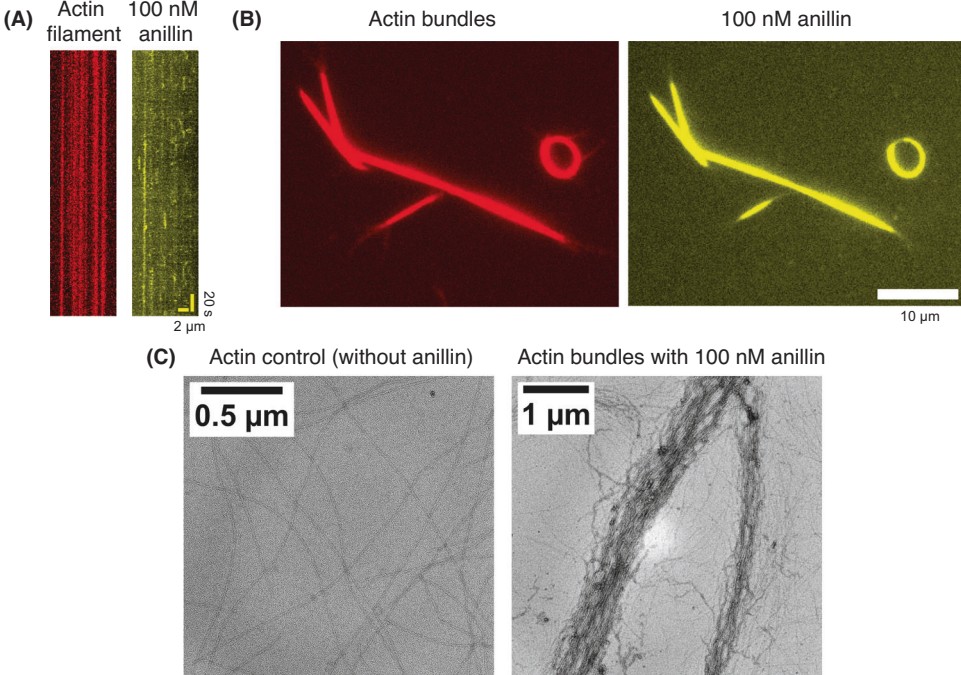

**Figure EV2.   Anillin forms heterogenous and loose actin bundles.**

(A) Kymograph of 100 nM anillin (yellow) bound to a single surface-immobilised actin filament (red). (B) TIRF images of actin bundles formed by 100 nM anillin. (C) EM images of actin filaments without (left; re-displayed from Fig. 3F here as a reference) and with 100 nM anillin (right). With anillin, clear bundling of actin filaments is observed, and these bundles are loose with different widths.

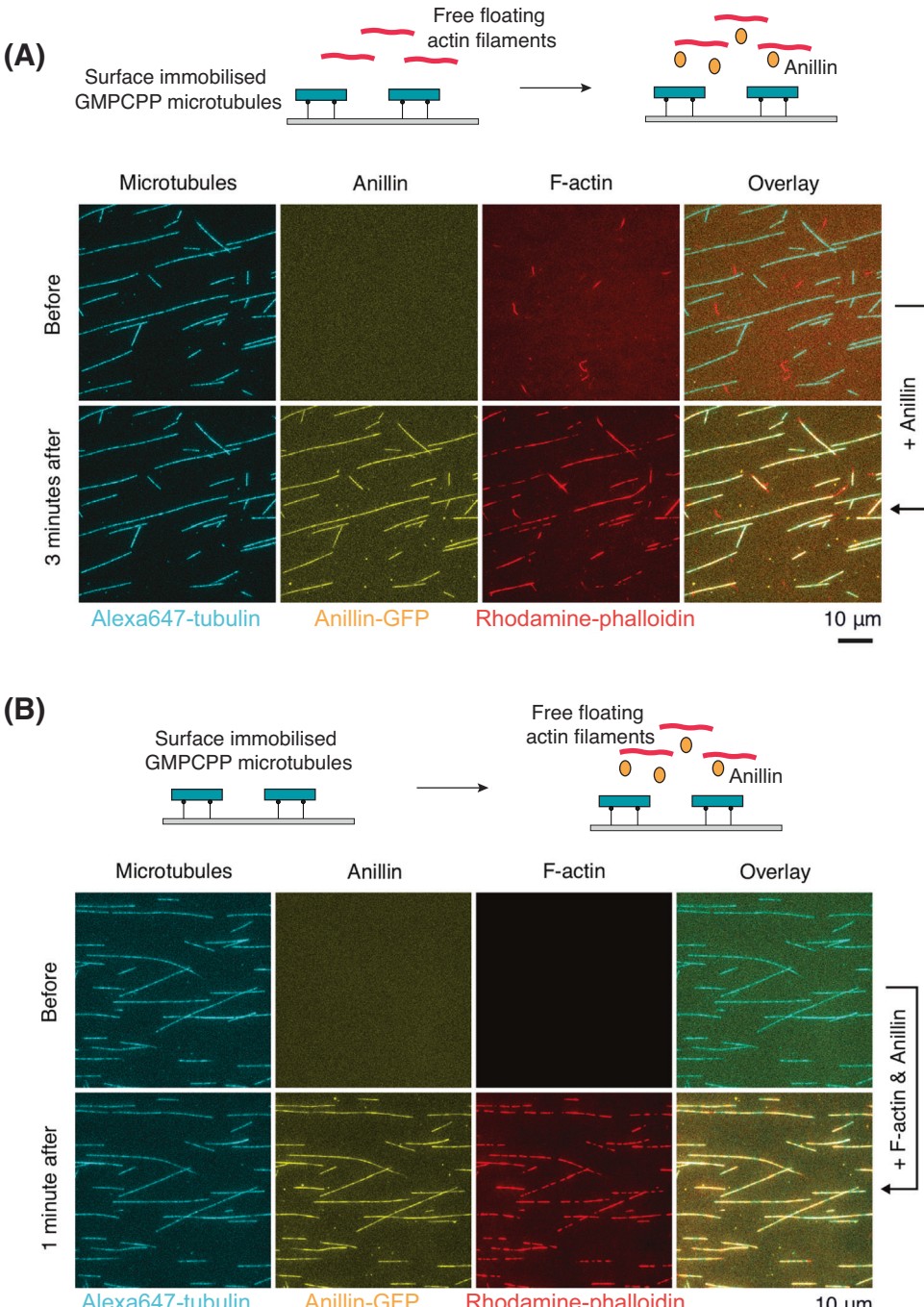

**Figure EV3. Crosslinking of actin filaments and microtubules by anillin occurs irrespective of the order in which components are combined.**

(A) Top: The flow channel initially contains surface-immobilised GMPCPP microtubules (cyan) and a solution of actin filaments (red) without anillin. Under these conditions, actin filaments freely diffuse and do not bind to the microtubules. Bottom: Once anillin-GFP (yellow) is flown in, actin filaments bind to the microtubules. (B) Top: The channel contains only surface-immobilised microtubules (cyan). Bottom: A solution of anillin-GFP (yellow) and actin filaments (red) is flown in, causing actin filaments to get bound to the microtubules. Sale bars 10 μm.

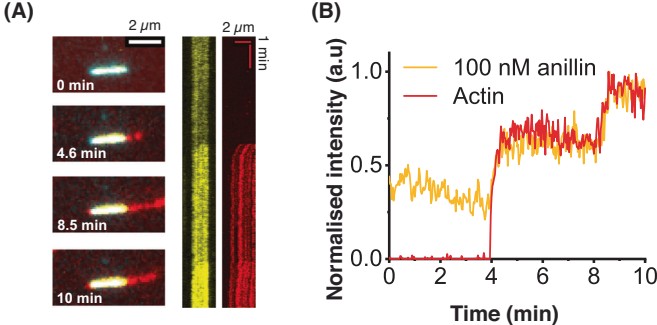

**Figure EV4.  Anillin crosslinks actin filaments and microtubules.**

Increase of the intensity of anillin (100 nM) upon the recruitment of actin filaments on a GMPCPP microtubule. The still images and kymograph (**A**) and corresponding fluorescence intensity plot (**B**) show that the anillin signal increases simultaneously with the recruitment of actin filaments.

