## [Peer Review File · The EMBO Journal]

Anillin directly cross-links microtubules with actin filaments

Ilna Bareja, Ondřej Kučera, Irene Petitjean, Beatriz Eugenia Orozco Monroy, Jan Sabo, Marcus Braun, Zdenek Lansky, Gijsje Koenderink, and Marileen Dogterom

Corresponding author(s): Marileen Dogterom (M.Dogterom@tudelft.nl) , Gijsje Koenderink (G.H.Koenderink@tudelft.nl)

Review Timeline:

Submission Date:	5th Jan 25
Editorial Decision:	10th Feb 25
Revision Received:	10th May 25
Editorial Decision:	20th May 25
Revision Received:	26th May 25
Accepted:	6th Jun 25

Editor: Hartmut Vodermaier

Transaction Report:

Prof. Marileen Dogterom
Technical University Delft
Department of Bionanoscience
Netherlands

10th Feb 2025

Re: EMBOJ-2024-120013
Anillin directly cross-links microtubules with filamentous actin

Dear Dr. Dogterom,

Thank you for submitting your study on actin/microtubule crosslinking by anillin to The EMBO Journal. We have now received the below-copied reports by three expert referees, who are all supportive of the work, its overall quality and its results. We would therefore be happy to consider a revised version further for publication, pending adequate addressing of a number of specific concerns and suggestions noted in the three reports. As you will see, the majority of these points pertain to data presentation/interpretation/discussion, but there are also a few experimental issues, most notably referee 2's request for deeper analyses in Figure 5.

It is our policy to consider only a single round of major revision and therefore important to fully respond to all comments at the time of resubmission; therefore, please do not hesitate to get back to me with a tentative response letter/revision plan, in case you would like to clarify/discuss certain points and how they might be answered already during the early stages of the revision. I should add that we could also offer extension of the default three-months revision period if needed, with our 'scooping protection' (meaning that competing work appearing elsewhere in the meantime will not affect our considerations of your study) remaining of course valid also throughout this extension.

Detailed information on preparing, formatting and uploading a revised manuscript can be found below and in our Guide to Authors. Thank you again for the opportunity to consider this work for The EMBO Journal, and I look forward to your revision in due time.

Yours sincerely,

Hartmut Vodermaier

4) Each main and each Expanded View (EV) figure should be uploaded as individual production-quality files (preferably in .eps,

.tif, .jpg formats). For suggestions on figure preparation/layout, please refer to our Figure Preparation Guidelines: <http://bit.ly/EMBOPressFigurePreparationGuideline>

9) To facilitate reproducibility and cross-laboratory adoption of methodologies, please structure the Materials & Methods section as outlined in our guide to authors, including a completed Reagents and Tools Table that can be downloaded from our author guidelines as well (<https://www.embopress.org/page/journal/14602075/authorguide#structuredmethods>).

10) Digital image enhancement is acceptable practice, as long as it accurately represents the original data and conforms to community standards. If a figure has been subjected to significant electronic manipulation, this must be clearly noted in the figure legend and/or the 'Materials and Methods' section. The editors reserve the right to request original versions of figures and the original images that were used to assemble the figure. Finally, we generally encourage uploading of numerical as well as gel/blot image source data; for details see: embopress.org/page/journal/14602075/authorguide#sourcedata

At EMBO Press, we ask authors to provide source data for the main manuscript figures. Our source data coordinator will contact you to discuss which figure panels we would need source data for and will also provide you with helpful tips on how to upload and organize the files.

In the interest of ensuring the conceptual advance provided by the work, we recommend submitting a revision within 3 months (11th May 2025). Please discuss the revision progress ahead of this time with the editor if you require more time to complete the revisions. Use the link below to submit your revision:

Link Not Available

Referee #1:

This well-written manuscript by Bareja et al. reveals a novel function for anillin as a crosslinker for actin and microtubule filament systems. Anillin is a key regulator of cytokinesis, a process that is critical for successful cell division, and has also been shown to control other cellular processes during development (e.g., at adhesion junctions). Cytokinesis occurs by the assembly and constriction of a RhoA-dependent ring in response to cues from the mitotic spindle. Anillin has binding domains for multiple regulators and components of the cell including RhoA, RhoA regulators, F-actin, active myosin, microtubules, septins and phospholipids. Through these many interactions, anillin has been described as a crosslinker for ring positioning. Anillin depletion in multiple cell types supports this function, as the ring can still form, but then oscillates around the cell and fails to ingress. The anaphase spindle has distinct pools of microtubules, including the central spindle, which arises between chromatin and consists of antiparallel bundles, and the centrosome-associated astral microtubules, which emanate toward the poles of the cell. Anillin is recruited to the equatorial cortex by active RhoA in anaphase. The N-terminus of anillin has an actin binding domain that was shown to bind and bundle F-actin, while the C-terminus of anillin has a binding domain for active RhoA, among other domains for binding to phospholipids and microtubules. Interestingly, anillin is not strongly localized to microtubules, but this localization increases when actin/myosin is decreased. In addition, microtubule-binding is autoregulated by the RhoA-binding domain. From prior studies, it was not clear how anillin binds to microtubules and whether it has a preference for the central spindle or astral microtubules, and what function this would have for cytokinesis. It was not clear if/how anillin could accommodate binding to both actin and microtubules simultaneously, and how binding to one or the other could impact the other system. This article is

important because it characterizes anillin's binding to microtubules, shows that it can bundle microtubules, and that it is possible for anillin to bind to both actin and microtubules, and can function as a crosslinker for the two systems. While this in vitro work does not answer the question of feedback per se, it brings us a step closer. Also, given that anillin has functions in cellular other processes, the findings from this manuscript could be informative for those in other fields.

I do not have major critiques of the manuscript - most of my suggestions are relatively minor.

Minor suggestions.

1. The Introduction nicely describes the importance and relevance of the work for the field. However, there are some minor referencing issues - just check to make sure that these align with the literature cited (e.g., Zhang and Maddox, 2010 or Hickson and O'Farrell).
2. In the Results, I appreciate that the authors justify their methodology, and refer to previously published studies of actin or microtubule proteins (binding, bundling), including anillin with F-actin. However, I would have liked to have seen a positive control included in their assays to give the reader a sense of how their approaches compare to what has been done before. For example, including PRC1 as a positive control for anillin in the microtubule-binding and bundling experiments.
3. In the Results, while the authors can identify the plus ends of microtubules in the kinetic/kymograph assays, I didn't get a sense of this in the assays for bundling microtubules. Especially given that the central spindle microtubules are antiparallel, do the authors know if there is a preference for anillin to bind or bundle microtubules in a parallel vs. antiparallel fashion?
4. In the schematic, the authors show us the different functions for anillin in binding and bundling actin and microtubules. This is a nice schematic for the reader. I would have liked to also have seen the inclusion of their data on how anillin binding to microtubule plus ends seems to increase their catastrophe and rescue (or perhaps I misunderstood this). What do the authors see as the role of anillin in microtubule stability and how does this fit into their idea of feedback between actin and microtubules?
5. In the Discussion, the authors speculate that the microtubule-actin crosslinking function of anillin could impact force generation, but I am struggling to see how that would work. Do they mean in the early stages as the actin filaments become more aligned and transition into the ring, which can cause cortical indentation? Or are they referring to force generation for constriction?
6. Also related to their Discussion, how do the authors see the crosslinking function of anillin changing during constriction when actin-myosin filaments are lost from the ring, and the central spindle microtubules are becoming highly condensed? I think it would be okay if the authors did not want to include any speculation of constriction (would take care of the comment above) and just speculated on earlier roles for ring positioning and organization given that the complexities of changes in both actin and microtubules during constriction.
7. Given the findings from Zanin's lab (Lebedev et al., 2023) showing that anillin can form independent linear structures - did the authors observe this in their system? Especially in the presence (or absence) of actin or microtubules? If the authors did or didn't, it would be good to include some reference to this article. I think the authors nicely explain how the disordered regions could gain structure via binding to actin or microtubules. Perhaps they could add a statement regarding whether they saw independent anillin structures/filaments.

Overall, I consider these suggestions to be minor, and if the authors choose not to repeat experiments using a well-characterized actin or MT-binding protein for comparison, then it would be good to include how they compare relatively based on published data (especially for microtubule binding and bundling).

Referee #2:

Please find enclosed the comments on the manuscript EMBOJ-2024-120013 entitled Anillin directly cross-links microtubules with filamentous actin.

The authors report the characterization of anillin interaction first with microtubules, then with actin filaments. Finally, they demonstrate that anillin is able to crosslink microtubules and actin filaments.

There are four main important aspects of this manuscript:

1. Anillin binds to microtubules (with a preference for GTP-subunits), and the kinetic parameters of this interaction allow anillin to diffuse along growing microtubules (or even tip-track).
2. Anillin appears to oligomerize in the presence of dynamic microtubules, and these oligomers are involved in microtubule bundling.

3. Anillin also binds to actin filaments and can mediate actin filament/microtubule interactions.

4. The diffusive capability of anillin along microtubules enables the transport (or contraction) of actin filaments along microtubules.

The data are presented in a remarkable way (the figures are beautifully constructed, maybe Figure 5 could be improved see comments below), and overall, they support the main conclusions of the paper.

However, I have three main comments that the authors should address to strengthen their manuscript before publication:

1. Mechanism of anillin oligomerization and microtubule bundling: The oligomerization of anillin is critical for understanding its role in microtubule bundling, actin filament bundling, and crosslinking between actin filaments and microtubules. While the authors demonstrate that oligomerization is favored in the presence of dynamic microtubules, they do not characterize the mechanism. Could free tubulin dimers be responsible for this oligomerization? In other words, does anillin bind to tubulin? If not, recent reports suggest that some microtubule-associated proteins (MAPs) undergo phase transition when interacting with microtubules. Could this be the case here? A deeper analysis of anillin oligomers along microtubules could help resolve these questions.

2. Dynamic actin filaments in crosslinking experiments: All experiments involving actin filaments are conducted using phalloidin-stabilized actin. While I understand the challenge of studying a system where both actin filaments and microtubules are dynamic, I believe the authors have the capability to support at least their main observation of crosslinking between actin filaments and microtubules using both dynamic actin filaments and microtubules. It would be particularly interesting to observe the growth of actin filament bundles along microtubules mediated by anillin and track the dynamic organization of these bundles over time. I encourage the authors to attempt these experiments. If significant technical challenges prevent them from doing so, they could address this limitation in their response to the reviewers.

3. The sliding of anillin along microtubules is a very interesting observation, allowing for some form of diffusive transport (without motors). If we extend this concept, the authors could observe some degree of contraction or packing of actin filament bundles along microtubules (Figure 5D). Perhaps the authors could expand their analysis of the data presented in Figure 5 to quantitatively illustrate the four behaviors they report upon crosslinking actin filaments with microtubules mediated by anillin. For example, regarding binding and unbinding, they could quantify the dependency of this behavior as a function of actin length and anillin intensity. So far, they have only presented a snapshot of a single experiment, which is a weakness of the manuscript.

Referee #3:

Summary:

The study "Anillin directly crosslinks microtubules with filamentous actin" by Barejia et al. presents strong evidence that the canonical actin-binding protein anillin also binds to microtubules and functions as a bona fide actin-microtubule crosslinker. This mechanistic study uses high-resolution *in vitro* assays to provide new insights into how essential cellular processes, such as division, are executed and how these mechanisms may apply to other cellular contexts (e.g., cell-cell junctions, transport). Specifically, the authors demonstrate anillin's activities by measuring several detailed parameters with actin filaments, microtubules, and both polymers simultaneously. They convincingly show that anillin exhibits a preference for higher-order cytoskeletal structures, such as bundles of actin or bundles of microtubules. Additionally, the study highlights anillin's role during actin-microtubule crosslinking events where there is a redistribution of anillin to these sites of actin-microtubule overlaps. These findings have significant implications for cytokinesis, and additional potential functions in other cellular contexts (i.e., cell-cell junctions, morphology). Overall, this is an excellent study that will be of great interest to cell biologists and cytoskeletal aficionados, providing valuable insights for the field.

Major concerns:

Anillin increases catastrophe frequency but also enhances rescue frequency, indicating a complex role in microtubule stability. This should be emphasized more in the text, as it suggests a dual function rather than pure destabilization. A small figure visualizing these data would be more accessible than a table.

Several figures are missing panels of relevant controls. To be clear the authors did perform them, but it would benefit the readers to have them in the same panels as the relevant test, rather than in the supplement. The specific examples are: 1) a microtubule alone (0nM anillin) control for Figure 1D; 2) Figure 2 FRAP analysis should specify why the median is used rather than the mean and include the sample size; 3) The EM images (Figure 3F and 4D) should include the control lacking anillin in a nearby panel as part of the main figure (they are available in the supplement). This would make the author's argument even more striking/obvious to the reader.

The argument that anillin prefers bundled filaments is well documented. However an example of anillin binding to single actin

filaments (before the TIRF field is full of bundles) would further strengthen this idea and help the reader understand what kind of data was measured for the comparison of single filaments and bundles later in the figure.

In Figure 4 the redistribution of anillin to sites of actin-microtubule overlap is so cool. This finding made this reviewer very curious about whether anillin had a preference for actin or microtubules and if that preference contributed to this effect. From going through the Koff data the values seem pretty close. Perhaps the authors can guide the readership to if they think anillin is specifically tipping the balance for one polymer or the other?

Minor concerns:

The statement in the introduction that anillin is "primarily nuclear during interphase" raises an intriguing question of whether it is performing a role with nuclear actin or transcription. This should be addressed.

Although I understand the author's intent, the claim that the full-length anillin ensures physiological relevance is overstated. Instead, the authors should clarify that they used full-length anillin rather than the truncated versions (studied in the past elsewhere) to better reflect the in vivo scenario.

The study exclusively uses a GFP-tagged version of anillin. Does the tag contribute to any of the invitro activities for actin or microtubule dynamics?

Anillin has been reported to phase separate in other contexts (i.e., DOI: 10.1021/acs.biochem.9b00217; DOI: 10.1038/s41467-025-56645-9). It might broaden the scope of the discussion (or potential readership) if the authors discuss whether the observed clustering could be linked to this property of anillin or a distinct mechanism.

A space is missing between "a-actinin" and the "Winkelman" reference in the discussion.

Non essential suggestions:

The authors could consider rewording the title to "Anilin directly crosslinks microtubules with actin filaments" which conveys the same meaning but is possibly accessible to a wider audience.

The authors could consider converting Table 1 into graphs as a small figure or additional panels to add to Figure 1.

Itemized response - Manuscript EMBOJ-2024-120013**Bareja et al., Anillin directly crosslinks microtubules with actin filaments****Referee #1:**

This well-written manuscript by Bareja et al. reveals a novel function for anillin as a crosslinker for actin and microtubule filament systems. Anillin is a key regulator of cytokinesis, a process that is critical for successful cell division, and has also been shown to control other cellular processes during development (e.g., at adhesion junctions). Cytokinesis occurs by the assembly and constriction of a RhoA-dependent ring in response to cues from the mitotic spindle. Anillin has binding domains for multiple regulators and components of the cell including RhoA, RhoA regulators, F-actin, active myosin, microtubules, septins and phospholipids. Through these many interactions, anillin has been described as a crosslinker for ring positioning. Anillin depletion in multiple cell types supports this function, as the ring can still form, but then oscillates around the cell and fails to ingress. The anaphase spindle has distinct pools of microtubules, including the central spindle, which arises between chromatin and consists of antiparallel bundles, and the centrosome-associated astral microtubules, which emanate toward the poles of the cell. Anillin is recruited to the equatorial cortex by active RhoA in anaphase. The N-terminus of anillin has an actin binding domain that was shown to bind and bundle F-actin, while the C-terminus of anillin has a binding domain for active RhoA, among other domains for binding to phospholipids and microtubules. Interestingly, anillin is not strongly localized to microtubules, but this localization increases when actin/myosin is decreased. In addition, microtubule-binding is autoregulated by the RhoA-binding domain. From prior studies, it was not clear how anillin binds to microtubules and whether it has a preference for the central spindle or astral microtubules, and what function this would have for cytokinesis. It was not clear if/how anillin could accommodate binding to both actin and microtubules simultaneously, and how binding to one or the other could impact the other system.

This article is important because it characterizes anillin's binding to microtubules, shows that it can bundle microtubules, and that it is possible for anillin to bind to both actin and microtubules, and can function as a crosslinker for the two systems. While this in vitro work does not answer the question of feedback per se, it brings us a step closer. Also, given that anillin has functions in cellular other processes, the findings from this manuscript could be informative for those in other fields. I do not have major critiques of the manuscript - most of my suggestions are relatively minor.

We thank the reviewer for the positive appraisal of the importance of our work and the suggestions for improving the manuscript, which we address point-by-point below.

Minor suggestions

1. The Introduction nicely describes the importance and relevance of the work for the field. However, there are some minor referencing issues - just check to make sure that these align with the literature cited (e.g., Zhang and Maddox, 2010 or Hickson and O'Farrell).

We thank the reviewer for pointing out this issue. We have double-checked all the references and corrected the bibliography.

2. In the Results, I appreciate that the authors justify their methodology, and refer to previously published studies of actin or microtubule proteins (binding, bundling), including anillin with F-actin. However, I would have liked to have seen a positive control included in their assays to give the reader a sense of how their approaches compare to what has been done before. For example, including PRC1 as a positive control for anillin in the microtubule-binding and bundling experiments.

We appreciate this suggestion but choose not to compare different crosslinkers directly in this study experimentally. We have instead included references in the discussion to address the comparison for microtubule bundling. We had previously included multiple references to compare the tip-tracking activity of anillin to other nonmotor MAPs, as well as its ability to recognize different microtubule lattices. We have now also added a reference comparing bundling by anillin to PRC1. Additionally, we also added a reference to compare actin-microtubule bundling by anillin to that of tau.

3. In the Results, while the authors can identify the plus ends of microtubules in the kinetic/kymograph assays, I didn't get a sense of this in the assays for bundling microtubules. Especially given that the central spindle microtubules are antiparallel, do the authors know if there is a preference for anillin to bind or bundle microtubules in a parallel vs. antiparallel fashion?

We have added this new experiment to the appendix (Appendix figure S2) and also mentioned this correspondingly in the Results section. Using IRM combined with TIRF microscopy we observed that this bundling of microtubules by anillin is polarity independent.

4. In the schematic, the authors show us the different functions for anillin in binding and bundling actin and microtubules. This is a nice schematic for the reader. I would have liked to also have seen the inclusion of their data on how anillin binding to microtubule plus ends seems to increase their catastrophe and rescue (or perhaps I misunderstood this). What do the authors see as the role of anillin in microtubule stability and how does this fit into their idea of feedback between actin and microtubules?

Prompted by the reviewer's suggestion, we moved the Table that summarizes the microtubule dynamics data to the Supplementary Material and included new panels in Figure 1 (panels L – O) showing the growth speeds, shrinkage speeds, catastrophes and rescues. The data indeed show that anillin may increase the catastrophe and rescue rates. Since the effect on the rescue rate is most pronounced and independent on whether anillin accumulates at the end or not, the overall effect seems to be a stabilizing one. We added a remark about this stabilizing effect in the discussion.

5. In the Discussion, the authors speculate that the microtubule-actin crosslinking function of anillin could impact force generation, but I am struggling to see how that would work. Do they mean in the early stages as the actin filaments become more aligned and transition into the ring, which can cause cortical indentation? Or are they referring to force generation for constriction?

This was indeed a highly speculative comment motivated by previous findings showing that crosslinkers such as anillin can generate contractile forces by a condensation mechanism. We deleted this comment from the revised manuscript.

6. Also related to their Discussion, how do the authors see the crosslinking function of anillin changing during constriction when actin-myosin filaments are lost from the ring, and the central spindle microtubules are becoming highly condensed? I think it would be okay if the authors did not want to include any speculation of constriction (would take care of the comment above) and just speculated on earlier roles for ring positioning and organization given that the complexities of changes in both actin and microtubules during constriction.

We now focus the discussion of anillin's potential role in cell division toward its involvement in ring positioning and have removed our speculative comments about the possible impact of anillin on ring constriction. We acknowledge there is insufficient understanding of the interplay between actin and microtubule cytoskeleton during constriction.

7. Given the findings from Zanin's lab (Lebedev et al., 2023) showing that anillin can form independent linear structures - did the authors observe this in their system? Especially in the presence (or absence) of actin or microtubules? If the authors did or didn't, it would be good to include some reference to this article. I think the authors nicely explain how the disordered regions could gain structure via binding to actin or microtubules. Perhaps they could add a statement regarding whether they saw independent anillin structures/filaments.

We did not observe such linear structures; we added a comment on this in the revised discussion section, with a reference to the Lebedev et al. paper.

Overall, I consider these suggestions to be minor, and if the authors choose not to repeat experiments using a well-characterized actin or MT-binding protein for comparison, then it would be good to include how they compare relatively based on published data (especially for microtubule binding and bundling).

Referee #2:

The authors report the characterization of anillin interaction first with microtubules, then with actin filaments. Finally, they demonstrate that anillin is able to crosslink microtubules and actin filaments. There are four main important aspects of this manuscript:

- 1. Anillin binds to microtubules (with a preference for GTP-subunits), and the kinetic parameters of this interaction allow anillin to diffuse along growing microtubules (or even tip-track).*
- 2. Anillin appears to oligomerize in the presence of dynamic microtubules, and these oligomers are involved in microtubule bundling.*
- 3. Anillin also binds to actin filaments and can mediate actin filament/microtubule interactions.*
- 4. The diffusive capability of anillin along microtubules enables the transport (or contraction) of actin filaments along microtubules.*

The data are presented in a remarkable way (the figures are beautifully constructed, maybe Figure 5 could be improved see comments below), and overall, they support the main conclusions of the paper.

However, I have three main comments that the authors should address to strengthen their manuscript before publication.

We thank the reviewer for the positive appraisal of the importance and clarity of our manuscript, and the constructive feedback, which we address point-by-point below.

Main comments

1. Mechanism of anillin oligomerization and microtubule bundling: The oligomerization of anillin is critical for understanding its role in microtubule bundling, actin filament bundling, and crosslinking between actin filaments and microtubules. While the authors demonstrate that oligomerization is favored in the presence of dynamic microtubules, they do not characterize the mechanism. Could free tubulin dimers be responsible for this oligomerization? In other words, does anillin bind to tubulin? If not, recent reports suggest that some microtubule-associated proteins (MAPs) undergo phase transition when interacting with microtubules. Could this be the case here? A deeper analysis of anillin oligomers along microtubules could help resolve these questions.

We checked for the direct interaction between free tubulin and anillin using iSCAT. We were limited by the concentration we could use since this technique does not work well to detect individual particles beyond a total of 100 nM of protein. At different ratios of anillin to tubulin (e.g. 60 nM anillin and 60 nM tubulin or 50 nM anillin and 40 nM tubulin) we did not detect a direct interaction and/or oligomers of anillin. However, this mechanism can still not be ruled out since the concentration of tubulin used is much higher (15 μ M) in the TIRF assays. We have added this as a new panel to supplementary figure (Appendix) 1, and discuss this in the results section. We also discuss about potential phase separation of anillin as a cause for these oligomers as well.

2. Dynamic actin filaments in crosslinking experiments: All experiments involving actin filaments are conducted using phalloidin-stabilized actin. While I understand the challenge of studying a system where both actin filaments and microtubules are dynamic, I believe the authors have the capability to support at least their main observation of crosslinking between actin filaments and microtubules using both dynamic actin filaments and microtubules. It would be particularly interesting to observe the growth of actin filament bundles along microtubules mediated by anillin and track the dynamic organization of these bundles over time. I encourage the authors to attempt these experiments. If significant technical challenges prevent them from doing so, they could address this limitation in their response to the reviewers.

We were indeed interested in experiments with dynamic actin filaments. Such experiments would require buffer conditions that include the presence of 50 mM KCl. Unfortunately, we found that in the presence of 50 mM KCl, anillin did not bind to dynamic microtubules.

3. The sliding of anillin along microtubules is a very interesting observation, allowing for some form of diffusive transport (without motors). If we extend this concept, the authors could observe some degree of contraction or packing of actin filament bundles along microtubules (Figure 5D). Perhaps the authors could expand their analysis of the data presented in Figure 5 to quantitatively illustrate the four behaviors they report upon crosslinking actin filaments with microtubules mediated by anillin. For example, regarding binding and unbinding, they could quantify the dependency of this behavior as a function of actin length and anillin intensity. So far, they have only presented a snapshot of a single experiment, which is a weakness of the manuscript.

Prompted by the reviewer's suggestion we added 2 new panels to Figure 5 showing quantification of the time spent by actin filaments (single versus bundled) on microtubules (Fig 5E) and statistics for the different anillin-mediated actin-microtubule interaction events (Fig 5F).

Referee #3:

The study “Anillin directly crosslinks microtubules with filamentous actin” by Barejia et al. presents strong evidence that the canonical actin-binding protein anillin also binds to microtubules and functions as a bona fide actin-microtubule crosslinker. This mechanistic study uses high-resolution in vitro assays to provide new insights into how essential cellular processes, such as division, are executed and how these mechanisms may apply to other cellular contexts (e.g., cell-cell junctions, transport). Specifically, the authors demonstrate anillin’s activities by measuring several detailed parameters with actin filaments, microtubules, and both polymers simultaneously. They convincingly show that anillin exhibits a preference for higher-order cytoskeletal structures, such as bundles of actin or bundles of microtubules. Additionally, the study highlights anillin’s role during actin-microtubule crosslinking events where there is a redistribution of anillin to these sites of actin-microtubule overlaps. These findings have significant implications for cytokinesis, and additional potential functions in other cellular contexts (i.e., cell-cell junctions, morphology).

Overall, this is an excellent study that will be of great interest to cell biologists and cytoskeletal aficionados, providing valuable insights for the field.

We thank the reviewer for the positive evaluation of the quality of our work and the thoughtful suggestions, which we followed up point-by-point as detailed below.

Major concerns:

1. Anillin increases catastrophe frequency but also enhances rescue frequency, indicating a complex role in microtubule stability. This should be emphasized more in the text, as it suggests a dual function rather than pure destabilization. A small figure visualizing these data would be more accessible than a table.

Prompted by the reviewer’s suggestion, we moved the Table that summarizes the microtubule dynamics data to the Supplementary Material and included new panels in Figure 1 (panels L – O) showing the growth speeds, shrinkage speeds, catastrophes and rescues.

2. Several figures are missing panels of relevant controls. To be clear the authors did perform them, but it would benefit the readers to have them in the same panels as the relevant test, rather than in the supplement. The specific examples are: 1) a microtubule alone (0nM anillin) control for Figure 1D; 2) Figure 2 FRAP analysis should specify why the median is used rather than the mean and include the sample size; 3) The EM images (Figure 3F and 4D) should include the control lacking anillin in a nearby panel as part of the main figure (they are available in the supplement). This would make the author's argument even more striking/obvious to the reader.

We added the requested control experiments and explanations. First, we included a panel for the microtubule-alone control in Figure 1C (right). Second, we added an explanation of the use of the median rather than mean in FRAP analysis in the revised Methods section and included the same size in all FRAP figures (Fig. 2D, 3E, and 4F). Third, we moved the

EM images of control samples lacking anillin from the supplementary information to the main text figures (Fig. 3F and 4D).

3. The argument that anillin prefers bundled filaments is well documented. However an example of anillin binding to single actin filaments (before the TIRF field is full of bundles) would further strengthen this idea and help the reader understand what kind of data was measured for the comparison of single filaments and bundles later in the figure.

A different example is shown now for this, wherein the 1st two images show just actin filaments and no bundling. The bundling process is then specifically pointed out with arrows.

4. In Figure 4 the redistribution of anillin to sites of actin-microtubule overlap is so cool. This finding made this reviewer very curious about whether anillin had a preference for actin or microtubules and if that preference contributed to this effect. From going through the K_{off} data the values seem pretty close. Perhaps the authors can guide the readership to if they think anillin is specifically tipping the balance for one polymer or the other?

The predicted intrinsically disordered regions of anillin suggest that electrostatic forces likely govern its interactions with binding partners, and thus are sensitive to the ionic composition of the buffer. Even though the k_{off} from both cytoskeletal components is indeed very close in our experiments, we may expect that direct comparison between actin and microtubules will likely be context-dependent and influenced by specific biochemical conditions, precluding broader conclusions. Nevertheless, in the context of our experiments, we show that for microtubules, anillin preferentially binds to GMPCPP- or GTP-stabilised lattices over GDP lattices. There is also a slight preference for GMPCPP microtubules over single actin filaments. Our assays consistently show that anillin prefers filament overlaps over single filaments.

Minor concerns:

1. The statement in the introduction that anillin is "primarily nuclear during interphase" raises an intriguing question of whether it is performing a role with nuclear actin or transcription. This should be addressed.

NOTE: a paper just came out showing that nuclear ANLN regulates transcription initiation related Pol II clustering and target gene expression: Nuclear ANLN regulates transcription initiation related Pol II clustering and target gene expression - PubMed

We thank the reviewer for alerting us to this interesting work, which we now mention in the Introduction.

2. Although I understand the author's intent, the claim that the full-length anillin ensures physiological relevance is overstated. Instead, the authors should clarify that they used full-length anillin rather than the truncated versions (studied in the past elsewhere) to better reflect the in vivo scenario.

We rephrased this passage of the Introduction according to the reviewer's suggestion.

3. The study exclusively uses a GFP-tagged version of anillin. Does the tag contribute to any of the in vitro activities for actin or microtubule dynamics?

We acknowledge the reviewer's point regarding the exclusive use of GFP-tagged anillin in our study. Given the intrinsically disordered nature of anillin, it is indeed possible that the GFP tag influences its conformation and, by extension, its in vitro activity on actin and microtubules. To assess whether the observed behaviour is specific to GFP, we conducted control experiments using anillin tagged with mCherry. These experiments, although not included in the current manuscript, yielded qualitatively identical results to those obtained with GFP-anillin, suggesting that the effects we report are not attributable to the specific tag used. That said, we agree that fully resolving the influence of fluorescent tags on anillin's activity would require a more comprehensive study, ideally comparing untagged, GFP-, and mCherry-tagged versions under matched conditions. This remains a valid question, but one that falls outside the scope of the present work. We also note that, despite these limitations, most of what is known about individual protein behaviour in reconstituted systems is based on tagged constructs. Fluorescent fusions remain essential for direct observation, even as they introduce potential caveats. We are mindful of this in our interpretation of the data.

4. Anillin has been reported to phase separate in other contexts (i.e., DOI: 10.1021/acs.biochem.9b00217; DOI: 10.1038/s41467-025-56645-9). It might broaden the scope of the discussion (or potential readership) if the authors discuss whether the observed clustering could be linked to this property of anillin or a distinct mechanism.

We indeed speculate in the Discussion that anillin's reported ability to phase separate could contribute to its clustering on dynamic microtubules.

5. A space is missing between "a-actinin" and the "Winkelman" reference in the discussion.

We corrected this typo.

Non essential suggestions:

1. The authors could consider rewording the title to "Anilin directly crosslinks microtubules with actin filaments" which conveys the same meaning but is possibly accessible to a wider audience.

We agree and have changed the title.

2. The authors could consider converting Table 1 into graphs as a small figure or additional panels to add to Figure 1.

The moved Table 1 to the Supplementary Information and added the data in additional panels of Figure 1.

Prof. Marileen Dogterom
Technical University Delft
Department of Bionanoscience
Netherlands

20th May 2025

Re: EMBOJ-2024-120013R
Anillin directly cross-links microtubules with actin filaments

Dear Dr. Dogterom,

Thank you for submitting your revised manuscript to The EMBO Journal. Two of the original referees have now assessed it once more, and I am happy to say that both were fully satisfied with the revisions and have no further concerns at this stage. After incorporation of the following remaining editorial issues, we should therefore be able to proceed with formal acceptance of the study:

- Please mark all (co-)corresponding authors and indicate their email contact addresses on the title page of the manuscript.
- Please double-check to make sure to all relevant funding information in the manuscript is congruent with the info entered into our submission system. Currently missing in the submission system are: institutional support from CAS (RVO: 86652036) and Imaging Methods Core Facility at BIOCEV supported by the MEYS CR [LM2023050, Czech-Biolmaging]
- Please include a Disclosure and competing interests statement (next to the Acknowledgment section) - for details, see <https://www.embopress.org/competing-interests>
- As we are switching from a free-text author contribution statement towards a more formal statement based on Contributor Role Taxonomy (CRediT) terms, please remove the present Author Contribution section and instead specify each author's contribution(s) directly in the Author Information page of our submission system during upload of the final manuscript. See <https://casrai.org/credit/> for more information.
- Please carefully go through the reference list and make sure that each reference is complete with citation year, volume, and page/locator numbers. I also note that there is a bioRxiv preprint listed in the reference list but not cited in the text. Please make sure to adjust its citation format as specified for preprints in our author guidelines:
The citation in the text should be: "(preprint: NAME1 et al, YEAR)"
The citation in the reference list: "Author NAME1, Author NAME2, ... (YEAR) article title. bioRxiv doi: XXX"
- Please double-check that all figure panels and other data items are referenced at least once in the text. Currently, call-outs for Fig. 1B; 2D; Appendix Table S1 appear to be missing.
- Please rename the supplementary movies as Expanded View movies (in-text callouts again "Movie EV1/2/..."). Their legends should be moved out of the text into individual text files, each of which should be combined with the respective movie file into a separate ZIP file and uploaded as such.
- Finally, during routine pre-acceptance checks, our data editors have raised the following queries regarding figures, data, and legends, which I would ask you to address (ideally using the Track Changes option):
 1. Please note that the exact p values are not provided in the legends of figures 1K-O; 4C, 5E
 2. Please note that information related to n is missing in the legends of figures 1L-O; 4C
 3. Please note that the error bars are not defined in the legend of figure 1K.
 4. Please note that the measure of center for the error bars needs to be defined in the legends of figures 1L-O

I am returning the manuscript to you for a final round of revision, solely to allow you to make these modifications and upload the revised files. Once we will have received them, we should be ready to swiftly proceed with formal acceptance and production of the manuscript.

With kind regards,

Hartmut

*** PLEASE NOTE: All revised manuscripts are subject to initial checks for completeness and adherence to our formatting guidelines. Revisions may be returned to the authors and delayed in their editorial re-evaluation if they fail to comply to the following requirements (see also our Guide to Authors for further information):

9) To facilitate reproducibility and cross-laboratory adoption of methodologies, please structure the Materials & Methods section as outlined in our guide to authors, including a completed Reagents and Tools Table that can be downloaded from our author guidelines as well (<https://www.embopress.org/page/journal/14602075/authorguide#structuredmethods>).

10) Digital image enhancement is acceptable practice, as long as it accurately represents the original data and conforms to community standards. If a figure has been subjected to significant electronic manipulation, this must be clearly noted in the figure legend and/or the 'Materials and Methods' section. The editors reserve the right to request original versions of figures and the original images that were used to assemble the figure. Finally, we generally encourage uploading of numerical as well as gel/blot image source data; for details see: embopress.org/page/journal/14602075/authorguide#sourcedata

In the interest of ensuring the conceptual advance provided by the work, we recommend submitting a revision within 3 months (18th Aug 2025). Please discuss the revision progress ahead of this time with the editor if you require more time to complete the

revisions. Use the link below to submit your revision:

Link Not Available

Referee #2:

In their revised manuscript, the authors have addressed my comments satisfactorily.

Referee #3:

The authors have adequately addressed all my concerns with this revision and I look forward to seeing this work in print!